# scPPDM: A Diffusion Model for Single-Cell Drug–Response Prediction

## Abstract

This paper introduces the Single-Cell Perturbation Prediction Diffusion Model (scP-PDM), the first diffusion-based framework for single-cell drug-response prediction from scRNA-seq data. scPPDM couples two condition channels, pre-perturbation state and drug with dose, in a unified latent space via non-concatenative GD-Attn. During inference, factorized classifier-free guidance exposes two interpretable controls for state preservation and drug-response strength and maps dose to guidance magnitude for tunable intensity. Evaluated on the Tahoe-100M benchmark under two stringent regimes, unseen covariate combinations (UC) and unseen drugs (UD), scPPDM sets new **state-of-the-art** results across log fold-change recovery, $\Delta$ correlations, explained variance, and DE-overlap. Representative gains include **+36.11%/+34.21%** on DEG logFC-Spearman/Pearson in UD over the second-best model. This control interface enables transparent what-if analyses and dose tuning, reducing experimental burden while preserving biological specificity.

## 1 Introduction

The integration of high-throughput single-cell RNA sequencing (scRNA-seq) (Klein et al., 2015; Macosko et al., 2015; Zheng et al., 2017) and perturbation screens (Datlinger et al., 2017; Srivatsan et al., 2020) offers the opportunity to systematically characterize transcriptional responses to the coupled effects of cellular context, drug, and dose at single-cell resolution. It provides a data resource for elucidating mechanisms of action (MoA) (Lamb et al., 2006; Subramanian et al., 2017; Trapotsi et al., 2022), evaluating candidate compounds (Ye et al., 2018; Corsello et al., 2020), and exploring precision therapeutic strategies. However, existing experimental paradigms face significant bottlenecks. While enabling parallel observation of drug effects at some scale, traditional high-throughput screening (HTS) workflows (Mayr & Bojanic, 2009; Macarron et al., 2011; Iversen et al., 2012) are extremely costly in terms of reagents, personnel time, and turnaround. Moreover, their throughput is limited, making it infeasible to cover all potential cell line $\times$ drug $\times$ dose combinations; even within large public perturbation atlases, the actually measured conditions constitute only a tiny fraction of the combinatorial space.

Prior work mainly follows three directions. Latent-space encoder–decoder methods learn a shared space for counterfactual prediction and compose cellular context (Lotfollahi et al., 2019; 2021); some also encode drug and dose and leverage molecular structure (Hetzel et al., 2022; Qi et al., 2024). Optimal transport (OT) formulates unpaired mapping from control to perturbed distributions (Bunne et al., 2023; Dong et al., 2023). In addition, large-scale Transformer pretraining (Cui et al., 2024; Adduri et al., 2025; Hao et al., 2024) yields transferable gene-expression representations for downstream prediction and generation.

**Model overview.** We present the Single-Cell Perturbation Prediction Diffusion Model (scPPDM). Diffusion operates entirely in latent space with time embeddings. Conditions are encoded as (i) baseline state $z_{\mathrm{pre}}$ from the shared encoder's posterior mean and (ii) a structure-aware drug vector fused with dose via FiLM to produce $\tilde{z}_{\mathrm{drug}}$. These are fused into a compact token and injected non-concatenatively within the denoiser via GD-Attn (Section 3.2). At inference, we form a decomposable guidance rule with two coefficients $(s_p, s_d)$ and modulate $s_d$ by dose. See Figure 1 for an overview.

Our main contributions are summarized as follows:

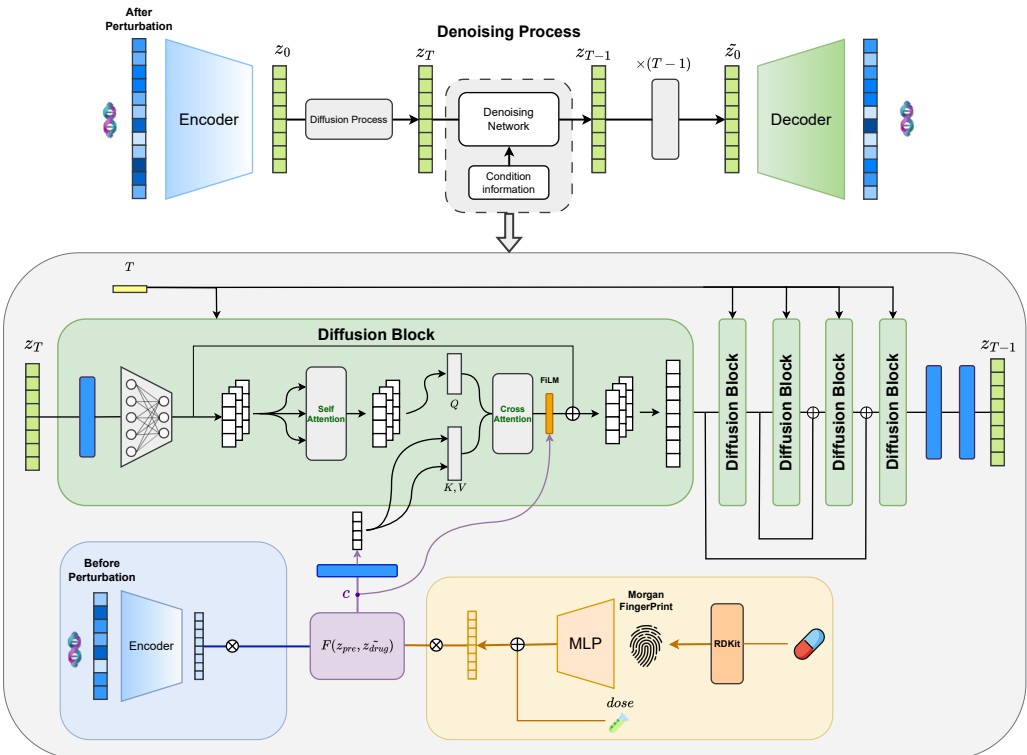

Figure 1: Overview of scPPDM. Top: In a shared VAE latent space, VP diffusion denoises $z_t$ (time-embedded) to $\hat{z}_0$ and decodes to $\hat{x}$. Bottom: Conditions comprise $z_{\text{pre}}=E(x_{\text{pre}})$ and a dose-fused drug vector $\tilde{z}_{\text{drug}}$ from Morgan fingerprints + MLP (Rogers & Hahn, 2010; Landrum et al.). They are fused into a token $c$ and injected non-concatenatively via GD-Attn (Section 3.2), yielding stable training and interpretable state/drug control.

- **Diffusion-based framework.** We are the first to apply a denoising diffusion model to predict post-perturbation expression at single-cell resolution, leveraging a unified latent space with strong learning and generative capacity for this task.

- **Dual-channel conditioning with dose-guided inference.** We are the first to model pre-perturbation state and drug as two separate condition channels and to control inference by mapping dose to drug-channel guidance strength, enabling transparent, decomposable guidance and interpretable intensity dialing.

**Results at a glance.** On the Tahoe-100M benchmark with UC/UD splits (UC: unseen cell line–condition; UD: unseen drugs), scPPDM outperforms linear and deep baselines across log fold-change recovery, $\Delta$ correlations, explained variance, and DE-overlap. Representative margins include **+13.46%** and **+12.00%** on DEG $\Delta$-Pearson and DEG logFC-Pearson in UC, and **+36.11%** and **+34.21%** on DEG logFC-Spearman and DEG logFC-Pearson in UD. On Top-1000 DEG-Accuracy, we further lead by **+21.88%** (UC) and **+28.57%** (UD), all relative to the second-best model.

## 2 RELATED WORK

**Deep learning for single-cell perturbation prediction.** Latent-space generative approaches perform counterfactual prediction by mapping expression into a shared latent space: scGen uses latent vector arithmetic across cell types (Lotfollahi et al., 2019); CPA represents cellular background and dose as additive components (Lotfollahi et al., 2021); chemCPA further incorporates molecular structure encoders and transfer from bulk HTS to handle previously unseen compounds at single-cell resolution (Hetzel et al., 2022; Subramanian et al., 2017). Structure-conditional generators

combine chemical structure (e.g., SMILES/fingerprints Weininger (1988); Schwartz et al. (2013)), dose, and control expression to predict responses for untested compounds (Qi et al., 2024). Large-scale Transformer pretraining learns transferable gene-expression representations that can serve as initialization or encoders for downstream prediction and generation (Cui et al., 2024). Another line of latent representation learning is exemplified by BioLORD (Piran et al., 2024), which trains cross-modal encoders linking molecular structure and transcriptomic responses, enabling generalization to previously unseen compounds; in contrast, SAMS-VAE (Bereket & Karaletsos, 2024) models perturbation effects only for the perturbations observed during training and does not support unseen-drug prediction. As an alternative to latent generative modeling, optimal-transport (OT) methods formulate control to perturbation alignment from unpaired observations to capture population-level shifts (Bunne et al., 2023; Dong et al., 2023). A hybrid line combines large Transformers with OT, exemplified by STATE (Adduri et al., 2025), though its generative formulation does not support prediction for previously unseen compounds.

**Diffusion and transferable techniques from computer vision.** In computer vision, diffusion provides an effective, scalable recipe for conditional generation: latent-space denoising with cross-attention (Rombach et al., 2022); controllable sampling via classifier or classifier-free guidance and compositional guidance (Ho & Salimans, 2022; Liu et al., 2022); Transformer backbones with feature-wise modulation (AdaLN/FiLM) (Vaswani et al., 2017; Peebles & Xie, 2023; Perez et al., 2018); and control adapters for selective layerwise conditioning (Zhang et al., 2023; Mou et al., 2023). More broadly, diffusion injects noise via a forward SDE and denoises via the reverse SDE or the probability-flow ODE (Song et al., 2020b), with accelerated non-Markovian sampling such as DDIM (Song et al., 2020a). For single-cell expression, scDiffusion (Luo et al., 2024) combines latent-space denoising with a pretrained encoder and uses classifier guidance and gradient-based conditional interpolation at inference. In experiments, the generated cells exhibit high fidelity to real data, preserving gene–gene correlation structure, cell type–specific expression patterns, and conditional consistency under perturbational settings.

**Data resource.** Large-scale perturbational resources enable compound–dose–cell-line coverage: LINCS/L1000 and related chemical encodings, and the recent Tahoe-100M (Zhang et al., 2025), a drug perturbation map of $>100\,\text{M}$ cells across 50 cell lines and $>1{,}100$ small molecules (about 60,000 drug–cell line combinations), which we use in this work.

## 3 METHOD

We organize scPPDM as a latent response schema with three coupled components. First, a fine-tuned SCimilarity VAE (Heimberg et al., 2023) defines the shared latent geometry and a denoising backbone learns response dynamics in Section 3.1. Second, baseline states and drug information are encoded, fused, and injected through Guided Decomposable Attention (GD-Attn) in Section 3.2. Third, training leverages four-state independent channel dropout, and inference applies factorized classifier-free guidance with controllable knobs $(s_p, s_d)$, including a dose-dependent schedule $s_d(\text{dose})$, as detailed in Section 3.3. An overview is provided in Figure 1.

### 3.1 LATENT-SPACE DIFFUSION BACKBONE

We model the post-perturbation expression distribution in a observational manner as the conditional mapping :

$$p(x_{\text{post}} \mid x_{\text{pre}}, d) \tag{1}$$

Let $x_{\text{pre}}, x_{\text{post}} \in \mathbb{R}^G$ denote the pre- and post-perturbation expression over $G$ genes, and let $d = (\text{SMILES}, \text{dose})$.

A fine-tuned SCimilarity VAE encoder $E$ and decoder $D$ define a latent space of dimension $D_z$ in which the response dynamics are learned. Once fine-tuned, $E$ and $D$ are frozen. Post-perturbation cells are mapped to $z_0 = E(x_{\text{post}})$ for training, and the decoder reconstructs predictions via $\hat{x} = D(\hat{z}_0)$ after denoising.

Within this latent schema, a variance-preserving diffusion process injects Gaussian noise along a schedule $\{\beta_t\}_{t=1}^T$ with $\alpha_t = 1 - \beta_t$ and cumulative product $\bar{\alpha}_t = \prod_{s \leq t} \alpha_s$. The forward transition is

$$q(z_t \mid z_0) = \mathcal{N}\big(\sqrt{\bar{\alpha}_t}\, z_0,\ (1 - \bar{\alpha}_t)\,\mathbf{I}\big). \tag{2}$$

The denoiser $B_\theta$ receives the noisy latent $z_t$ together with a sinusoidal time embedding $\gamma(t) \in \mathbb{R}^{d_t}$ and predicts the Gaussian noise $\hat{\epsilon}_t = B_\theta(z_t, \gamma(t))$. Conditions are injected later via GD-Attn (Section 3.2) rather than input concatenation, which we found to amplify batch noise and harm generalization in early training.

## 3.2 Condition Representation and GD-Attn Injection

We encode the pre-perturbation state with the shared VAE encoder and build a dose-aware, structure-informed drug vector before fusing both through GD-Attn.

**State encoding.** We reuse the same fine-tuned VAE encoder $E$ of the backbone on the baseline state conditioning branch.

$$z_{\text{pre}} = E^{\text{cond}}(x_{\text{pre}}) \in \mathbb{R}^{D_{\text{pre}}}, \quad D_{\text{pre}} = D_z \tag{3}$$

$z_{\text{pre}}$ is the conditioning-side latent for the baseline state; $D_{\text{pre}}$ matches the backbone latent dimension $D_z$; $E^{\text{cond}} \equiv E$ ("$\equiv$" indicates full parameter sharing).

To ensure stability and reproducibility, we use the encoder's posterior mean rather than sampling. In an ablation, we find introducing a separate condition-side encoder produces coordinate misalignment and direction drift during guidance.

$$z_{\text{pre}}^{\text{cond}} = \mathbb{E}_{q_\phi(z|x_{\text{pre}})}[z] = \mu_\phi(x_{\text{pre}}) \tag{4}$$

$\mu_\phi(\cdot)$ is the encoder's mean head.

**Drug encoding.** We let structure determine the perturbation direction and dose determine the magnitude. SMILES are converted to 1024-bit Morgan fingerprints $\phi \in \{0, 1\}^{1024}$ and embedded via an MLP to obtain the structure-aware vector

$$z_{\text{drug}} = \text{MLP}(\phi) \in \mathbb{R}^{D_{\text{drug}}} \tag{5}$$

where $D_{\text{drug}}$ is the dimensionality of the learned drug embedding. Dose is fused through a FiLM module with a continuous encoding $e_{\text{dose}} \in \mathbb{R}^{D_{\text{dose}}}$, where $D_{\text{dose}}$ denotes the dose-embedding dimension.

$$\tilde{z}_{\text{drug}} = \text{FiLM}\left(z_{\text{drug}}, e_{\text{dose}}\right) \tag{6}$$

Here FiLM is a learnable feature-wise linear modulation that alters the scale and shift of $z_{\text{drug}}$ without rotating its direction. Directly concatenating dose to $z_{\text{drug}}$ degrades directional stability in our tests while FiLM preserves the separation between direction and magnitude.

**Condition injection.** Guided Decomposable Attention (GD-Attn) performs the non-concatenative injection. The composite condition is

$$c = F\left(z_{\text{pre}}, \tilde{z}_{\text{drug}}\right) \in \mathbb{R}^{D_c} \tag{7}$$

$F$ is the fusion function and $D_c$ is the dimensionality of the condition.

GD-Attn (i) performs self-attention to refine latent representations, (ii) executes cross-attention against the compact condition token to couple state/drug signals, and (iii) applies FiLM modulation followed by a residual connection and LayerNorm. Implementation details and full equations are provided in Appendix G.

This non-concatenative pathway mitigates early-training batch-noise amplification and gradient-scale issues, preserves individualized baselines, and enables decomposable guidance at inference with separate state/drug channels.

## 3.3 Four-State Training and Factorized Guidance

We train scPPDM with four-state dropout (Section 3.3.1). At inference, we use linear, decomposable guidance with two knobs—$s_p$ (state) and $s_d$ (drug), with mapping dose $\rightarrow s_d(\text{dose})$ for intensity control (Section 3.3.2).

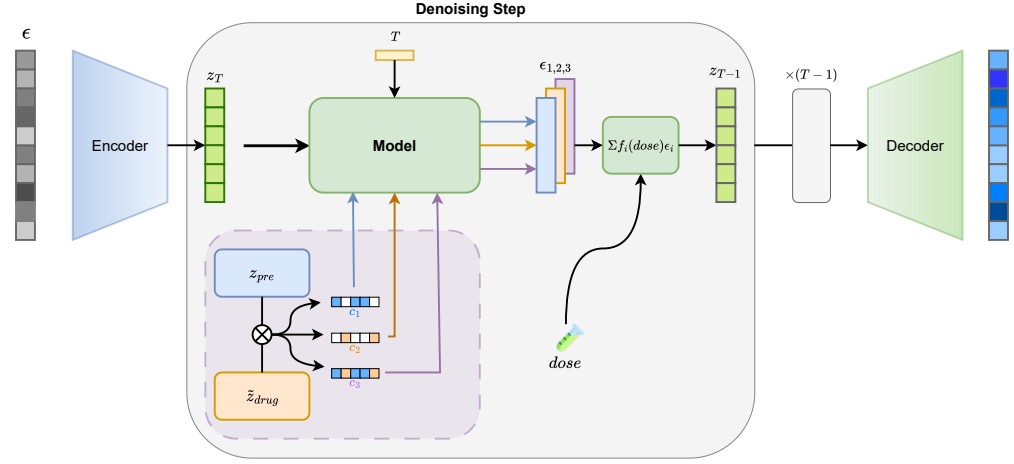

Figure 2: Decomposable, dose-controlled inference. At denoising step $t$, the model produces $\epsilon$ under both/no-pre/no-drug and forms $\hat{\epsilon}_t = (1 + s_p + s_d)\epsilon_{\text{both}} - s_p\epsilon_{\text{no-pre}} - s_d\epsilon_{\text{no-drug}}$, where $s_d$ is mapped from dose. The combined guidance updates $z_t \to z_{t-1}$; decoding $D(\hat{z}_0)$ gives the predicted post-perturbation $\hat{x}$.

### 3.3.1 FOUR-STATE TRAINING

Four-state training provides sufficient primitives for decomposable composition at inference, enabling robust control even on unseen pairings or unseen drugs. We apply independent dropout to $\{z_{\text{pre}}, \tilde{z}_{\text{drug}}\}$ in the training process to obtain the four possible combinations.

$$c' \in \{\text{both, no-pre, no-drug, } \varnothing\} \tag{8}$$

Unified training objective is shown below.

**Injection-side alignment and contrast.** For each sample $i$, define the normalized empirical response direction

$$\hat{v}^{(i)} = \frac{x_{\text{post}}^{(i)} - x_{\text{pre}}^{(i)}}{\|x_{\text{post}}^{(i)} - x_{\text{pre}}^{(i)}\|_2}. \tag{9}$$

We project the dose-fused embedding $\tilde{z}_{\text{drug}}$ with a learnable matrix $W$ and align it to $\hat{v}^{(i)}$ while maintaining contrast with negatives. The alignment loss is

$$\mathcal{L}_{\text{align}} = \left\| \hat{v}^{(i)} - \frac{W\,\tilde{z}_{\text{drug}}}{\|W\,\tilde{z}_{\text{drug}}\|_2} \right\|_2^2. \tag{10}$$

To preserve separability across drugs, we add an InfoNCE term (van den Oord et al., 2018; Chen et al., 2020):

$$\mathcal{L}_{\text{info}} = \mathbb{E}\left[ -\log \frac{\exp\big(\langle W\,\tilde{z}_{\text{drug}},\, \hat{v}^{(i)} \rangle / \tau\big)}{\sum_{v' \in \mathcal{N}} \exp\big(\langle W\,\tilde{z}_{\text{drug}},\, v' \rangle / \tau\big)} \right], \tag{11}$$

where $\langle \cdot, \cdot \rangle$ is the inner product, $\tau$ is the temperature, and $\mathcal{N}$ is the negative set. The injection regularizer combines them as

$$\mathcal{L}_{\text{inject}} = \lambda_{\text{align}}\,\mathcal{L}_{\text{align}} + \lambda_{\text{info}}\,\mathcal{L}_{\text{info}} \tag{12}$$

**Denoising loss under four-state training.** With independent channel dropout $c'$, the diffusion denoiser is trained by the MSE on noise:

$$\mathcal{L}_{\text{denoise}} = \mathbb{E}_{z_0,\, t,\, \epsilon,\, c'}\left[ \big\| \epsilon - \epsilon_\theta(z_t, t, c') \big\|_2^2 \right] \tag{13}$$

where $z_t \sim q(z_t \mid z_0)$ follows the VP forward process and $\epsilon_\theta$ predicts the noise.

**Total objective.** Our overall objective weights the denoising criterion with the injection regularizer:

$$\mathcal{L}_{\text{total}} = \mathcal{L}_{\text{denoise}} + \mathcal{L}_{\text{inject}} \tag{14}$$

with $\lambda_{\text{align}}, \lambda_{\text{info}} \geq 0$ as hyperparameters.

### 3.3.2 FACTORIZED GUIDANCE

The inference scheme is provided by Figure 2.

**Dual-Knob Mechanism.** We set two knobs with distinct roles: $s_p$ preserves individualized baseline consistency, while $s_d$ amplifies the drug direction. Linear composability is justified by the score–noise identity under VP diffusion; see Appendix A.2.1.

$$\hat{\epsilon} = (1 + s_p + s_d)\,\epsilon(\text{both}) - s_p\,\epsilon(\text{no-pre}) - s_d\,\epsilon(\text{no-drug}), \qquad s_p, s_d \geq 0 \tag{15}$$

$s_p$ and $s_d$ respectively tune the guidance strengths of the state and drug channels to yield the final denoising direction; $\epsilon(\cdot)$ denotes the network output under the indicated condition. In practice, we find that moderately increasing $s_p$ helps preserve individualized baselines when extrapolating across cell lines.

**Dose-to-Strength Mapping.** We apply dose information to better control $s_d$ at inference.

$$s_d(\text{dose}) = s_0 \cdot \sigma\big(\alpha \log(1 + d) + \gamma\big), \qquad \sigma(z) = \frac{1}{1 + e^{-z}} \tag{16}$$

Using log-dose (Sebaugh, 2011) with a sigmoid controls the guidance strength of the drug channel and is continuous at zero dose; $d$ is the drug dosage; $s_0, \alpha, \gamma$ are hyperparameters. We initially use a linear mapping, which causes over-saturation at high doses. The sigmoid fits the dose–response shape more stably and aligns with classical dose–response curves in pharmacology.

## 4 EXPERIMENT

We first fine-tune SCimilarity VAE (Heimberg et al., 2023) on the training set to accommodate the gene dimension of our dataset, then freeze the encoder/decoder and perform all diffusion training and inference within this latent space. We take scDiffusion (Luo et al., 2024) as the architectural reference and tailor it to our setting. Optimization uses AdamW (Loshchilov & Hutter, 2017); the learning rate warms up for 3,000 steps and then decays linearly to a small value over training progress. Detailed experimental settings and hyperparameters are provided in Appendix D.

### 4.1 BENCHMARK

#### 4.1.1 DATASET

We use the Tahoe-100M (Zhang et al., 2025) public single-cell drug-perturbation atlas as our data source. This dataset is at hundred-million scale, spans 50 cancer cell lines, includes 379 drugs in its public release, and is compatible with the conditional mapping $(x_{\text{pre}}, \text{drug}, \text{dose}) \rightarrow x_{\text{post}}$ that we aim to learn. We take $(\text{cell\_line}, \text{drug}, \text{dose}, \text{plate})$ as conditioning keys, averaging single-cell expression within each key group to obtain the corresponding representative state. For each non-DMSO condition, we identify the matched DMSO control on the same plate to establish a one-to-one pair, yielding the pre-perturbation transcriptome $x_{\text{pre}}$ and post-perturbation transcriptome $x_{\text{post}}$. Using these samples as the atomic unit, we obtain splits for training and inference. In the tests we use both UC (unseen covariate combinations: cell lines and drugs seen individually during training, but whose combination is unseen) and UD (unseen drugs: test-set drugs never appear in training), ensuring no key overlap and no data leakage. A formal rationale for not directly adopting random single-cell pairing nor its variants is provided in Appendix E.

#### 4.1.2 METRICS

We rank all genes by absolute $\log_2$ fold change ($|\log_2 \text{FC}|$) under the perturbation and take the top-$k$ genes as the high-impact set. For all DEG-based evaluations except (4), we fix $k = 200$ and treat this set as the differentially expressed genes (DEGs). We evaluate the following metrics (each computed on both the full gene set and the DEG subset): (1) Pearson and Spearman correlations between the

Table 1: UC test set results. Comparison of baseline models and our method on the UC scenario. Metrics are reported with ↑ indicating that higher is better. **Bold** and underline indicate the best and second method for each metrics, respectively.

| *(unseen combo)* | Ours | chemCPA | PRnet | STATE | scGPT | Linear | MLP | Perturb | Context | Biolord | SAMS-VAE |
|---|---|---|---|---|---|---|---|---|---|---|---|
| logFC-Pearson ↑ | **0.81** | 0.70 | 0.61 | 0.77 | 0.46 | 0.50 | 0.55 | 0.46 | 0.48 | 0.50 | 0.54 |
| logFC-Spearman ↑ | **0.69** | 0.46 | 0.58 | **0.69** | 0.39 | 0.51 | 0.57 | 0.38 | 0.32 | 0.34 | 0.57 |
| logFC-Pearson(DEG) ↑ | **0.56** | 0.40 | 0.31 | 0.50 | 0.32 | 0.38 | 0.37 | 0.35 | 0.35 | 0.43 | 0.44 |
| logFC-Spearman(DEG) ↑ | **0.49** | 0.38 | 0.28 | 0.48 | 0.23 | 0.34 | 0.23 | 0.33 | 0.32 | 0.29 | 0.45 |
| Δ-Pearson ↑ | 0.55 | **0.59** | 0.35 | 0.50 | 0.39 | 0.48 | 0.44 | 0.21 | 0.25 | 0.46 | 0.53 |
| Δ-Spearman ↑ | **0.51** | 0.44 | 0.27 | 0.50 | 0.35 | 0.41 | 0.36 | 0.25 | 0.24 | 0.37 | 0.55 |
| Δ-Pearson(DEG) ↑ | **0.57** | 0.52 | 0.36 | 0.54 | 0.38 | 0.45 | 0.43 | 0.22 | 0.24 | 0.38 | 0.46 |
| Δ-Spearman(DEG) ↑ | **0.59** | 0.46 | 0.28 | 0.52 | 0.33 | 0.41 | 0.36 | 0.24 | 0.25 | 0.31 | 0.49 |
| DEG-accuracy(top200) ↑ | **0.24** | 0.22 | 0.11 | 0.22 | 0.11 | 0.14 | 0.17 | 0.16 | 0.13 | 0.21 | 0.20 |
| DEG-accuracy(top1000) ↑ | **0.39** | 0.28 | 0.21 | 0.32 | 0.19 | 0.19 | 0.18 | 0.17 | 0.15 | 0.30 | 0.29 |
| *EV*_median ↑ | **0.73** | 0.70 | 0.51 | 0.69 | 0.41 | 0.20 | 0.40 | 0.48 | 0.39 | 0.63 | 0.53 |
| *EV*_median(DEG) ↑ | **0.64** | 0.59 | 0.37 | 0.53 | 0.34 | 0.18 | 0.30 | 0.38 | 0.33 | 0.51 | 0.44 |

ground-truth and predicted log fold-change vectors ($\log_2$ FC); (2) Pearson and Spearman correlations between the ground-truth and predicted perturbation-shift vectors ($\Delta := x_{post} - x_{pre}$); (3) Median explained variance of the predictions with respect to the ground truth; (4) DE-Overlap-Accuracy. For this task, we selected several k values for evaluation. Details are shown in Appendix C. (5) Perturbation Discrimination Score (PDS) (Liu et al., 2025).

To ensure that this effect-size–based selection reflects true biological signal, we performed per-gene paired t-tests followed by Benjamini–Hochberg FDR correction, and found that among the Top-200 genes, **84% (UC)** and **79% (UD)** satisfy FDR $\leq 0.05$.

### 4.1.3 BASELINES

Our model and other models are trained on the same set and evaluated on UC and UD test sets. All models use the same split, and each is trained to convergence. Baselines include Linear Regression and Multi-layer Perceptron (MLP) with drug encodings implemented analogously to PRnet (Qi et al., 2024) adapter. The Perturb (Adduri et al., 2025) baseline averages observed perturbation offsets for the same perturbation in training data and adds mean offset to control expression to form prediction. The Context (Adduri et al., 2025) baseline averages perturbed expression across samples within the same cell line/context and uses this mean as prediction directly. We further compare four deep-learning baselines: chemCPA (Hetzel et al., 2022), an additive latent-space model with molecular structure encoding and dose scaling, supporting tests on both UC and UD; PRnet (Qi et al., 2024), an encoder–decoder model conditioned on drug perturbations via adapter that generates fingerprint-based embeddings, supporting both tasks above; BioLORD (Piran et al., 2024) trains cross-modal encoders linking molecular structure and transcriptomic profiles and therefore supports both UC and UD evaluation; SAMS-VAE (Bereket & Karaletsos, 2024) models perturbation labels as discrete conditions without molecular-structure encoding and thus supports UC evaluation but cannot generalize to unseen drugs and is not evaluated on UD; STATE (Adduri et al., 2025), a model that takes unperturbed transcriptome together with the perturbation label, evaluated only on UC and not supporting UD prediction; and scGPT (Cui et al., 2024), a Transformer-based single-cell foundation model used as perturbation-prediction baseline, also evaluated only on UC without supporting UD prediction. All models, including baseline methods, are trained and evaluated on the same pseudobulked condition-level pairs for fairness.

## 4.2 RESULTS

### 4.2.1 COMPARISON WITH BASELINES

Across both UC and UD settings, our model achieves consistent **state-of-the-art (SoTA)** performance against baselines (Table 1, Table 2). Below we report several substantial improvements. On UC, we observe clear gains on DEG-focused correlation metrics, including **+13.46%** on DEG Δ-Pearson and **+12.00%** on DEG logFC-Pearson, both relative to the second-best method. On UD, which probes unseen-compound generalization, the margins become larger: **+36.11%** on DEG logFC-Spearman and **+34.21%** on DEG logFC-Pearson. Beyond correlations, our model also improves the discrete target-recovery measure Top-1000 DEG-Accuracy by **+21.88%** on UC and **+28.57%** on UD.

Table 2: UD test set results. Evaluation of models on the UD scenario. $\overline{\text{Perturb}}$, STATE, scGPT and SAMS-VAE are not included, as they do not directly support UD tasks. Metrics are reported with ↑ indicating that higher is better. **Bold** and underline indicate the best and second method for each metrics, respectively. For consistency, metric definitions follow those used in the UC scenario (Tables 1–2).

| (unseen drug) | Ours | Chemcpa | Prnet | Linear | MLP | Context | Biolord |
|---|---|---|---|---|---|---|---|
| logFC-Pearson ↑ | **0.67** | 0.42 | 0.59 | 0.43 | 0.40 | 0.31 | 0.42 |
| logFC-Spearman ↑ | **0.58** | 0.44 | 0.48 | 0.49 | 0.36 | 0.32 | 0.30 |
| logFC-Pearson(DEG) ↑ | **0.51** | 0.38 | 0.30 | 0.32 | 0.29 | 0.31 | 0.30 |
| logFC-Spearman(DEG) ↑ | **0.49** | 0.36 | 0.28 | 0.26 | 0.26 | 0.32 | 0.26 |
| Δ-Pearson ↑ | 0.50 | **0.53** | 0.33 | 0.46 | 0.43 | 0.42 | 0.45 |
| Δ-Spearman ↑ | **0.49** | 0.43 | 0.25 | 0.40 | 0.34 | 0.39 | 0.37 |
| Δ-Pearson(DEG) ↑ | **0.53** | 0.48 | 0.34 | 0.40 | 0.41 | 0.43 | 0.29 |
| Δ-Spearman(DEG) ↑ | **0.52** | 0.41 | 0.28 | 0.37 | 0.33 | 0.39 | 0.27 |
| DEGaccuracy(top200) ↑ | **0.22** | 0.20 | 0.07 | 0.12 | 0.14 | 0.15 | 0.19 |
| DEGaccuracy(top1000) ↑ | **0.36** | 0.28 | 0.19 | 0.17 | 0.16 | 0.15 | 0.29 |
| $EV$_median ↑ | **0.66** | 0.59 | 0.42 | 0.18 | 0.32 | 0.40 | 0.60 |
| $EV$_median(DEG) ↑ | **0.56** | 0.47 | 0.31 | 0.11 | 0.25 | 0.30 | 0.51 |

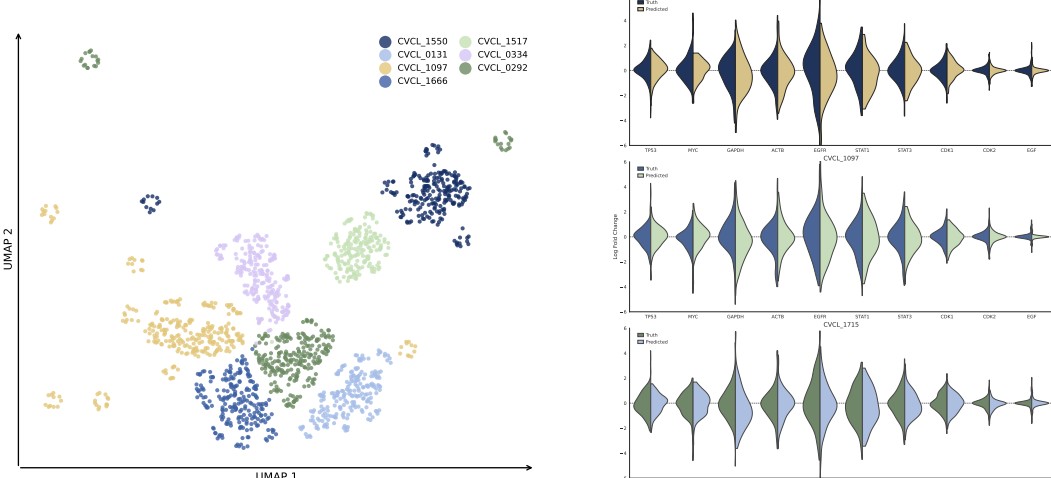

Figure 3: Visualization of predictions. Left: UMAP of model-predicted post-perturbation transcriptomes colored by cell line, showing that clusters align with cell-line identity. Right: Violin plots for CVCL-0069/1097/1715 showing gene-wise logFC distributions (Predicted vs. Truth), indicating strong agreement in direction and magnitude.

These results indicate that our design not only better aligns the direction of perturbation effects (reflected by Δ and logFC correlations), but also preserves salient targets under ranking-based selection (DEG-Accuracy). We further note that improvements are stable across metrics and data regimes: while a few baselines occasionally edge out specific metrics, such cases are isolated and not systematic. In contrast, our method exhibits broad, robust gains—particularly in the more challenging UD split.

Complete comparison tables can be found in Appendix I.

### 4.2.2 VISUALIZATION

We project model-predicted post-perturbation transcriptomes into the low-dimensional UMAP (McInnes et al., 2018) space. We can see that the predicted expression profiles from different cell lines can naturally cluster, clearly separating cell lines. This indicates the model retains cell-line-specific

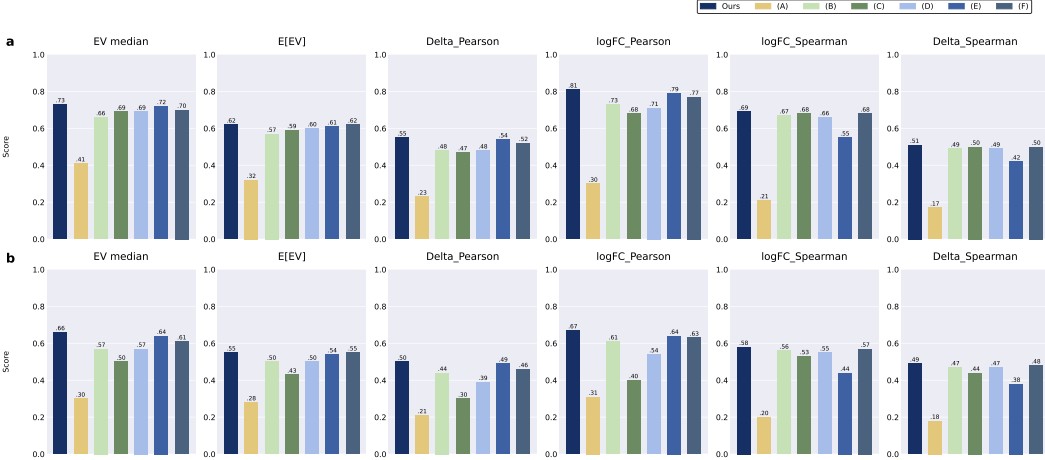

Figure 4: Ablations on **UC** (a) and **UD** (b). We evaluate five factors (A–E; see text). Trends are consistent: conditioning is required, GD-Attn (Section 3.2) and dual-knob guidance improve performance, structure priors enable UD generalization, and mapping dose$\rightarrow s_d$ preserves cross-dose monotonicity.

transcriptional signatures, demonstrating biological interpretability, and capturing drug-perturbation effects in cross–cell-line generalization. (Figure 3)

For specified cell lines (e.g., CVCL-0069, CVCL-1097, CVCL-1715) under various drug perturbation circumstances, violin plots display distributions of logFC for several relatively significantly changed genes from model predictions and ground-truth observations. Predicted and empirical distributions were highly consistent, indicating that the model accurately captured gene-level perturbation direction and magnitude. (Figure 3)

### 4.3 ABLATION STUDY

We ablate five factors by removing or replacing modules as follows: (A) remove all conditioning (No-Cond); (B) replace GD-Attn (Section 3.2) with Concat+MLP; (C) replace the Morgan/SMILES-based drug encoder (Equation (5)) with Drug-ID; (D) replace decomposable dual-knob guidance $(s_p, s_d)$ (Equation (15)) with single-channel CFG; and (E) replace the dose-to-strength mapping $s_d(\text{dose})$ (Equation (16)) with a constant. (F) replace the non-concatenative GD-Attn injection with an input-level concatenation before attention ("Concat+Attn"), which concatenates the fused condition token with the latent tokens prior to the attention blocks.

The ablation results are shown in Figure 4. Across UC/UD, we find that:

- Conditioning is indispensable—removing all conditions collapses performance;
- Non-concatenative injection (GD-Attn) improves fit and $\Delta$-alignment over Concat+MLP while preserving dose-rank behavior;
- Structure-aware priors are critical for UD generalization—replacing structure with Drug-ID hurts most;
- Decomposable guidance with dual knobs $(s_p, s_d)$ is more robust and interpretable than single-channel CFG;
- Mapping dose$\rightarrow s_d$ maintains cross-dose monotonicity, whereas a constant $s_d$ breaks it.

The complete ablation experiments description and further analysis are given in Appendix F.

## 5 DISCUSSION AND CONCLUSION

scPPDM is the first diffusion-based framework for single-cell perturbation prediction, built on three pillars: a unified latent-space backbone, non-concatenative conditioning, and factorized guidance. On

Tahoe-100M UC/UD splits it achieves consistent **SoTA** gains, and UMAP/violin analyses show biologically coherent, cell line–specific predictions. Ablations confirm the necessity of each component, including conditioning, GD-Attn, structure-aware drug encoding, dual-knob guidance, and dose $\to s_d$ mapping.

Looking ahead, scPPDM is an interpretable engine for rapid compound and dose scoring that shrinks wet-lab effort. Its dual knobs enable transparent what-if analyses and adaptation to patient/cohort baselines, supporting early-stage drug discovery and screening, as well as precision medicine.

## REPRODUCIBILITY STATEMENT

We have taken multiple steps to facilitate reproducibility. The model and training procedure are fully specified in Section 3 (latent VP diffusion in Section 3.1, condition encoding and GD-Attn injection in Section 3.2, four-state training and factorized guidance in Section 3.3), with extended theoretical details and identities provided in Appendix A (PF-ODE view, Tweedie relation, conditional score identity, DDIM error bound) and the guidance decomposition analysis in Appendix B. The dataset, pairing protocol, and UC/UD split construction are described in Section 4.1.1, with the rationale against random single-cell pairing in Appendix E. All evaluation metrics are precisely defined in Appendix C, including logFC/$\Delta$ correlations, explained variance, and DE-Overlap-Accuracy, along with aggregation rules. Full hyperparameters, schedules, optimization settings, and architecture specifics (e.g., DrugMapper, FiLM, guidance mapping) are listed in Appendix D; implementation details for tokenization and GD-Attn are in Appendix G. We report comprehensive ablations and settings in Appendix F, and baseline configurations/splits are summarized in Section 4.1.3. We prepare to release our experimental code to further support community reuse and extension upon acceptance.

## THE USE OF LARGE LANGUAGE MODELS (LLMs)

We use large language models (LLMs) to assist with translation and editing for grammar and style.

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

# A    EXTENDED THEORY FOR LATENT-SPACE VP DIFFUSION (SEC. 3.1)

## A.1    VECTOR-FIELD VIEW AND PF-ODE CONSISTENCY

We abstract the net drug effect on the baseline transcriptome by a vector field:

$$x_{\text{post}} = x_{\text{pre}} + g(x_{\text{pre}}, d) + \zeta, \quad \mathbb{E}[\zeta] = 0. \tag{17}$$

Training operates in the latent space $z$. Under the probability-flow ODE (PF-ODE) of a variance-preserving (VP) forward process, a convenient drift form is

$$\dot{z} = f_t(z) := -\tfrac{1}{2}\beta_t z - \tfrac{1}{2}\beta_t s_t(z),$$

where $s_t(z) = \nabla_z \log p_t(z \mid \cdot)$ is the (Gaussian-smoothed) conditional score. Deterministic DDIM ($\eta{=}0$) is a numerical integrator of this PF-ODE, whereas stochastic DDPM discretizes the reverse-time SDE.

## A.2    TWEEDIE RELATION AND POSTERIOR-MEAN CONSISTENCY

The Tweedie identity (Efron, 2011) links the smoothed score to the denoised posterior mean:

$$\mathbb{E}[z_0 \mid z_t] = \frac{1}{\sqrt{\bar{\alpha}_t}}\Big(z_t + \sigma_t^2 \, \nabla_z \log p_t(z_t)\Big), \tag{18}$$

with $\bar{\alpha}_t = \prod_{s \le t} \alpha_s$, $\alpha_t = 1 - \beta_t$, and $\sigma_t^2 = 1 - \bar{\alpha}_t$. Equivalently, if the network predicts noise $\widehat{\epsilon}_\theta(z_t)$, then

$$\widehat{z}_0 = \frac{1}{\sqrt{\bar{\alpha}_t}}\Big(z_t - \sigma_t \, \widehat{\epsilon}_\theta(z_t)\Big).$$

### A.2.1 CONDITIONAL SCORE IDENTITY & LINEAR DECOMPOSABILITY

Under the VP forward process, fixing $t$ and the condition $c$, the MSE-optimal solution is

$$\epsilon_\theta^\star(z_t, t, c) = \mathbb{E}[\epsilon \mid z_t, c] = -\sigma_t \nabla_z \log p_t(z_t \mid c), \qquad \nabla_z \log p_t(z_t \mid c) = -\frac{1}{\sigma_t} \epsilon_\theta^\star(z_t, t, c). \quad (19)$$

Here $\sigma_t = \sqrt{1 - \bar{\alpha}_t}$; $p_t$ is the smoothed distribution obtained by convolving with $\mathcal{N}(0, \sigma_t^2 \mathbf{I})$; $\nabla_z \log p_t$ is the smoothed conditional score. Let $c_{\text{both}} = (c_{\text{pre}}, c_{\text{drug}})$, $c_{\text{no-pre}} = (\varnothing, c_{\text{drug}})$, and $c_{\text{no-drug}} = (c_{\text{pre}}, \varnothing)$ for subsequent compositions. This identity links noise regression to the conditional score and thereby justifies the linear, decomposable composition used during inference, preserving individualized baselines while amplifying drug-driven directions.

### A.3 MULTI-SCALE NOISE AND HIERARCHICAL ALIGNMENT

By scheduling $\{\sigma_t\}$ from coarse to fine, the model first aggregates pathway-level signals and then refines gene-level differences, enabling pathway→gene hierarchical alignment that improves interpretability and robustness.

### A.4 PF-ODE DISCRETIZATION CONSISTENCY AND AN EXPLICIT ERROR BOUND FOR DDIM

Assume: (A1) $s_t(z)$ is $L$-Lipschitz in $z$ uniformly in $t$; (A2) $\beta_t \in [0, \beta_{\max}]$ and $\partial_t \beta_t$ is bounded; (A3) $\partial_t s_t(z)$ is uniformly bounded so that $\sup_{t,z} \|\partial_t f_t(z)\| \leq B$. Under (A1–A3), $f_t$ is $L_f$-Lipschitz in $z$ with

$$L_f \leq \tfrac{1}{2} \beta_{\max} (1 + L).$$

Let $z(t)$ be the PF-ODE solution with $z(0) = z_0$, and let $z_k^{\text{DDIM}}$ be the deterministic DDIM ($\eta=0$) iterate obtained by a one-step explicit integrator of step size $\Delta t$ over $t \in [0, 1]$ (so $k\Delta t = t_k$). Then the global error of explicit first-order integration satisfies the standard bound

$$\max_{0 \leq k \leq 1/\Delta t} \left\| z_k^{\text{DDIM}} - z(t_k) \right\|_2 \leq \frac{\Delta t}{2} \frac{B}{L_f} \left( e^{L_f} - 1 \right) = O(\Delta t). \quad (20)$$

In particular, there exists a constant $C > 0$ (depending on $L_f, B$) such that

$$\sup_{t \in [0,1]} \left\| z_{\text{DDIM}}(t) - z_{\text{PF}}(t) \right\|_2 \leq C \Delta t. \quad (21)$$

### A.5 A SUFFICIENT CONDITION FOR DIRECTIONAL CONSISTENCY

When a dominant mechanism of action (MoA) holds without strong alternative pathways, the drug-direction signal aligns with the conditional score:

$$\left\langle g(x_{\text{pre}}, d), \nabla_x \log p(x \mid x_{\text{pre}}, d) \right\rangle \geq \rho \|g\|_2 \|\nabla_x \log p\|_2, \qquad \rho > 0. \quad (22)$$

### A.6 PATHWAY SUBSPACES AND INTERPRETABLE PROJECTIONS

Let $\mathcal{P}$ index pathway/gene sets, with projections $\{\Pi_k\}_{k \in \mathcal{P}}$ mapping output changes into pathway subspaces. These pair naturally with dose-to-strength control to test monotonicity and saturation.

### A.7 OCCUPANCY APPROXIMATION AND CHEMO–BIO BRIDGING

Approximating receptor occupancy–effect curves by a log-dose sigmoid implements "structure sets direction; dose sets magnitude," matching our injection and guidance-strength design.

## B DECOMPOSABLE GUIDANCE: RESIDUALS AND APPROXIMATION ERRORS

### B.1 DEFINITION OF THE DECOMPOSED GUIDANCE FIELD $\hat{s}_t$

We work in the latent space. With factorized guidance strengths $s_p$ (pre-state) and $s_d(\text{dose})$ (drug channel), define

$$\hat{s}_t(z) = \underbrace{\nabla_z \log p_t(z)}_{\text{base}} + s_p \underbrace{\nabla_z \log p_t(c_{\text{pre}} \mid z)}_{\text{state}} + s_d(\text{dose}) \underbrace{\nabla_z \log p_t(c_{\text{drug}} \mid z)}_{\text{drug}}. \quad (23)$$

This mirrors the additive score identity

$$\nabla_z \log p_t(z \mid c_{\mathrm{pre}}, c_{\mathrm{drug}}) = \nabla_z \log p_t(z) + \nabla_z \log p_t(c_{\mathrm{pre}} \mid z) + \nabla_z \log p_t(c_{\mathrm{drug}} \mid z) + r_t(z),$$

up to the interaction residual $r_t(z)$.

## B.2 Interaction residual (latent-space form)

Define the interaction residual (with Gaussian smoothing at level $t$):

$$r_t(z) = \nabla_z \log p_t(c_{\mathrm{pre}}, c_{\mathrm{drug}} \mid z) - \nabla_z \log p_t(c_{\mathrm{pre}} \mid z) - \nabla_z \log p_t(c_{\mathrm{drug}} \mid z). \quad (24)$$

Smaller $r_t$ implies more precise linear decomposability of guidance.

## B.3 Bounded-interaction assumption and approximation bounds

If $\|r_t(z)\|_2 \le \lambda(t)$ almost everywhere, then

$$\left\| \hat{s}_t(z) - \nabla_z \log p_t(z \mid c_{\mathrm{pre}}, c_{\mathrm{drug}}) \right\|_2 \le \lambda(t), \qquad \left\| \hat{\epsilon}_t - \epsilon_t^\star(c_{\mathrm{both}}) \right\|_2 \le \sigma_t \lambda(t), \quad (25)$$

where $\epsilon_t^\star(c_{\mathrm{both}})$ denotes the Bayes-optimal noise predictor under the joint condition.

## C Metrics – Definitions and Interpretation

### C.1 Notation and Scope

For each condition key $c$ (e.g., a specific combination of cell line, drug, dose, plate), let $\mu_{\mathrm{post},g}^{(c)}$ and $\mu_{\mathrm{pre},g}^{(c)}$ denote the aggregated (e.g., plate-matched) perturbed and control means of gene $g \in \{1, \dots, G\}$. Unless otherwise stated, metrics are computed per condition over genes and then aggregated across conditions (we report mean/median in tables; the reducer does not affect the per-condition formulas below). A small pseudo-count $\varepsilon > 0$ is used in all log-ratios for numerical stability and is kept fixed across methods and splits.

### C.2 DEG Selection by Top-$K$ Absolute Log Fold Change

The condition-level log fold change (logFC) is

$$\mathrm{LFC}_g^{(c)} = \log_2 \frac{\mu_{\mathrm{post},g}^{(c)} + \varepsilon}{\mu_{\mathrm{pre},g}^{(c)} + \varepsilon}.$$

We define the DEG set as the top-$K$ genes by absolute magnitude,

$$S_c^{(K)} = \mathrm{TopK}_g \ |\mathrm{LFC}_g^{(c)}|.$$

By default we use $K{=}200$ for DEG-restricted correlation/EV metrics unless otherwise noted.

**What it measures.** This selects high-impact genes for condition $c$ independent of model predictions. Using ground truth for selection avoids "winner's curse" and ensures that DEG-restricted metrics stress recovery of salient targets.

### C.3 logFC Correlations (All Genes; DEG-Restricted)

Given predicted perturbed means $\widehat{\mu}_{\mathrm{post},g}^{(c)}$, we compute predicted logFC with the same control denominator:

$$\widehat{\mathrm{LFC}}_g^{(c)} = \log_2 \frac{\widehat{\mu}_{\mathrm{post},g}^{(c)} + \varepsilon}{\mu_{\mathrm{pre},g}^{(c)} + \varepsilon}.$$

Per condition $c$, Pearson and Spearman correlations over genes are

$$\rho_{\mathrm{P}}^{(c)} = \mathrm{Pearson}\left(\mathrm{LFC}_\bullet^{(c)}, \widehat{\mathrm{LFC}}_\bullet^{(c)}\right), \qquad \rho_{\mathrm{S}}^{(c)} = \mathrm{Spearman}\left(\mathrm{LFC}_\bullet^{(c)}, \widehat{\mathrm{LFC}}_\bullet^{(c)}\right).$$

DEG-restricted variants evaluate the same correlations on $S_c^{(K)}$:

$$\rho_{\text{P,DE}}^{(c)} = \text{Pearson}\big(\text{LFC}_{S_c^{(K)}}^{(c)}, \widehat{\text{LFC}}_{S_c^{(K)}}^{(c)}\big), \quad \rho_{\text{S,DE}}^{(c)} = \text{Spearman}\big(\text{LFC}_{S_c^{(K)}}^{(c)}, \widehat{\text{LFC}}_{S_c^{(K)}}^{(c)}\big).$$

**What they measure.** Pearson emphasizes magnitude fidelity (linear agreement), while Spearman emphasizes rank/order (robust to monotone rescaling). DEG-restricted scores focus on the biologically most perturbed genes.

## C.4 $\Delta$ (Post–Pre Shift) Correlations

Define condition-level mean shifts,

$$\overline{\Delta}_g^{(c)} = \mu_{\text{post},g}^{(c)} - \mu_{\text{pre},g}^{(c)}, \qquad \overline{\widehat{\Delta}}_g^{(c)} = \widehat{\mu}_{\text{post},g}^{(c)} - \mu_{\text{pre},g}^{(c)}.$$

Per condition,

$$\rho_{\Delta,\text{P}}^{(c)} = \text{Pearson}\big(\overline{\Delta}_\bullet^{(c)}, \overline{\widehat{\Delta}}_\bullet^{(c)}\big), \qquad \rho_{\Delta,\text{S}}^{(c)} = \text{Spearman}\big(\overline{\Delta}_\bullet^{(c)}, \overline{\widehat{\Delta}}_\bullet^{(c)}\big),$$

and their DEG-restricted counterparts replace "$\bullet$" by $S_c^{(K)}$. **Why $\Delta$ in addition to logFC.** $\Delta$ reflects absolute changes in the original scale (after any normalization), which is sensitive to baseline expression and complements logFC's relative-change view. Together they test both direction and scale consistency.

## C.5 Explained Variance ($E[r^2]$)

Let $\mathbf{y}^{(c)} \in \mathbb{R}^G$ be the ground-truth perturbed vector and $\hat{\mathbf{y}}^{(c)}$ its prediction. The per-condition explained variance is

$$\text{EV}^{(c)} = 1 - \frac{\text{Var}\big(\mathbf{y}^{(c)} - \hat{\mathbf{y}}^{(c)}\big)}{\text{Var}\big(\mathbf{y}^{(c)}\big)}.$$

**What it measures.** A scale-aware goodness-of-fit: 1 means perfect prediction; 0 means no better than predicting the mean of $\mathbf{y}^{(c)}$; negative values indicate worse-than-mean performance. Unlike correlations, EV penalizes global offsets and variance mis-calibration.

## C.6 DE-Overlap Accuracy (@$K$)

Let $S_c^{(K)}$ be the ground-truth Top-$K$ set by $|\text{LFC}^{(c)}|$ and $\widehat{S}_c^{(K)}$ be the predicted Top-$K$ by $|\widehat{\text{LFC}}^{(c)}|$.

$$\text{Acc}_{\text{DEG}}^{(c)} = \frac{|S_c^{(K)} \cap \widehat{S}_c^{(K)}|}{K}.$$

**What it measures.** Target-recovery at a fixed discovery budget (set overlap). We report $K{=}1000$ for this metric unless otherwise noted. (DEG-restricted correlations/EV use $K{=}200$ by default.)

## C.7 Aggregation Across Conditions, Reporting, and Robustness

Per-condition scores are aggregated across $c$ (UC/UD splits) using the median (robust) or mean (sensitive to extremes); the reducer used is stated in each table. For correlations, we also report DEG-restricted variants (Top-$K$) alongside all-gene variants to disentangle global calibration from salient-target recovery. When comparing across doses, Spearman (rank) is preferred for monotonicity checks; Pearson emphasizes dose-wise magnitude fidelity.

## C.8 Practical Notes and Pitfalls

- **Control denominator.** For both logFC and $\Delta$, the control $\mu_{\text{pre},g}^{(c)}$ in the denominator (or difference) is always the ground-truth control to avoid error coupling between control and post predictions.
- **Pseudo-count $\varepsilon$.** Use a single, fixed $\varepsilon$ across all methods/splits; varying $\varepsilon$ can alter logFC scales and confound comparisons, especially for lowly expressed genes.

- **Plate/context matching.** Perturbed–control pairing must respect plate/batch keys to avoid inflating apparent performance via leakage across contexts.
- **DEG choice.** Reporting both all-gene and DEG-restricted metrics prevents models from optimizing only the head or only the tail of the distribution.

## D  HYPERPARAMETERS AND TRAINING SETUP

**Latent space and backbone.**   Diffusion is trained and run entirely in the frozen **SCimilarity**-VAE (Heimberg et al., 2023) latent space; latent dimension $D_z = 128$.

**Four-state decomposed training.**   Independent masking on $\{z_{\mathrm{pre}}, \tilde{z}_{\mathrm{drug}}\}$: $p(\text{drop-pre}) = 0.10$, $p(\text{drop-drug}) = 0.10$. From independence,

$$P(\text{both}) = (1 - 0.10)(1 - 0.10) = 0.81,$$
$$P(\text{no-pre}) = 0.10(1 - 0.10) = 0.09,$$
$$P(\text{no-drug}) = (1 - 0.10)0.10 = 0.09,$$
$$P(\varnothing) = 0.10 \times 0.10 = 0.01.$$

Global dropout in the network is $0.15$.

**Noise schedule and sampling.**   VP-linear noise schedule. Inference uses DDIM.

**Factorized guidance.**   $s_p = 1.0$. Dose-to-strength mapping:

$$s_d(\text{dose}) = s_0 \cdot \sigma\big(\alpha \log(1 + \text{dose}) + \gamma\big), \quad s_0 = 3.0, \ \alpha = 2.0, \ \gamma = -0.5, \ \sigma(z) = \frac{1}{1+e^{-z}}.$$

**Loss weighting.**   $\mathcal{L}_{\mathrm{denoise}} : 1.0$, $\lambda_{\mathrm{align}} = 0.10$, $\lambda_{\mathrm{info}} = 0.05$; InfoNCE temperature $\tau = 0.1$. $\lambda_{\mathrm{align}}$ and $\lambda_{\mathrm{info}}$ linearly warm up to their targets during the first $20\%$ of training steps.

**Optimization and LR schedule.**   AdamW with learning rate $1.0 \times 10^{-4}$, weight decay $1.0 \times 10^{-5}$, $\beta_1 = 0.9$, $\beta_2 = 0.95$. Linear decay schedule: warm up for $3{,}000$ steps, then decay linearly to zero over training.

**Hardware and batch.**   We use $4 \times 80$ GB GPUs (data parallel). Per-GPU batch 256 (no grad accumulation), global batch 1024. FP16 mixed precision and gradient clipping (1.0) are enabled; EMA 0.999 for evaluation only.

**Drug representation and dose modulation.**   **DrugMapper** MLP: $1024 \rightarrow 512 \rightarrow 256 \rightarrow 128$, dropout $0.15$ between layers; output dim 128. Dose is fused via FiLM (hidden 128) and combined with $z_{\mathrm{drug}}$ in a residual manner.

**InfoNCE details.**   Temperature $\tau = 0.1$; 1024 negatives per step; memory bank length 16,384.

**Preprocessing and stability.**   Library-size normalization followed by $\log 1p$ on expression matrices. SMILES $\rightarrow$ 1024-bit Morgan fingerprints (radius 2).

## E  WHY WE DON'T USE RANDOM SINGLE-CELL PAIRING OR ITS VARIANTS

### E.1  SETTING AND NOTATION

Let a condition-group key be $G = (c, d, \delta, p)$, where $c$ denotes the cell line, $d$ the drug, $\delta$ the dose, and $p$ the plate. Denote pre-perturbation and post-perturbation single-cell expression by $X_{\mathrm{pre}}, X_{\mathrm{post}} \in \mathbb{R}^G$. For a fixed $G$,

$$\mu_{\mathrm{pre}}(G) = \mathbb{E}[X_{\mathrm{pre}} \mid G], \qquad \mu_{\mathrm{post}}(G) = \mathbb{E}[X_{\mathrm{post}} \mid G], \qquad \Sigma_{\mathrm{post}}(G) = \mathrm{Var}(X_{\mathrm{post}} \mid G).$$

A random pairing scheme draws two independent samples from the same group:

$$X_{\text{pre}}^{\text{rnd}} \sim P(\cdot \mid G), \quad X_{\text{post}}^{\text{rnd}} \sim P(\cdot \mid G), \quad X_{\text{post}}^{\text{rnd}} \perp\!\!\!\perp X_{\text{pre}}^{\text{rnd}} \mid G.$$

Training then minimizes the squared loss

$$\mathcal{R}(f) = \mathbb{E}\big[\|f(X_{\text{pre}}^{\text{rnd}}) - X_{\text{post}}^{\text{rnd}}\|_2^2\big],$$

with expectation over $G$ and conditional sampling.

### E.2 Consequence Under Squared Loss: Group-Mean Predictor

By the law of total expectation and the conditional independence above,

$$\mathcal{R}(f) = \mathbb{E}\Big[\|f(X_{\text{pre}}^{\text{rnd}}) - \mu_{\text{post}}(G)\|_2^2\Big] + \mathbb{E}\Big[\text{Var}(X_{\text{post}}^{\text{rnd}} \mid G)\Big].$$

The second term is independent of $f$. Hence the Bayes-optimal predictor within each group is

$$f^\star(x) = \mu_{\text{post}}(G),$$

i.e., a constant mapping that does not depend on the input $x$. Under random pairing, the optimal model collapses to the group-level post mean, thereby precluding the learning of cell-level dependencies.

### E.3 Gradient Noise Induced by Independence

For a differentiable parameterization $f_\theta$, the per-sample gradient is

$$g(\theta) = 2\big(f_\theta(X_{\text{pre}}^{\text{rnd}}) - X_{\text{post}}^{\text{rnd}}\big)\nabla_\theta f_\theta(X_{\text{pre}}^{\text{rnd}}).$$

Adding and subtracting $\mu_{\text{post}}(G)$ yields

$$g(\theta) = 2\Big(f_\theta(X_{\text{pre}}^{\text{rnd}}) - \mu_{\text{post}}(G)\Big)\nabla_\theta f_\theta(X_{\text{pre}}^{\text{rnd}}) - 2\Big(X_{\text{post}}^{\text{rnd}} - \mu_{\text{post}}(G)\Big)\nabla_\theta f_\theta(X_{\text{pre}}^{\text{rnd}}).$$

Conditioned on $G$ and $X_{\text{pre}}^{\text{rnd}}$, the second term has zero mean, while its conditional second moment scales as

$$\mathbb{E}\big[\|\eta\|_2^2 \mid G, X_{\text{pre}}^{\text{rnd}}\big] \propto \text{tr}\big(\Sigma_{\text{post}}(G)\big) \cdot \|\nabla_\theta f_\theta(X_{\text{pre}}^{\text{rnd}})\|_F^2, \qquad \eta \triangleq -2\big(X_{\text{post}}^{\text{rnd}} - \mu_{\text{post}}(G)\big)\nabla_\theta f_\theta(\cdot).$$

Thus random pairing injects an input-agnostic variance term into the gradient noise whose magnitude is governed by within-group variability $\Sigma_{\text{post}}(G)$, rather than informative cell-level coupling. For a sample size $N$, the variance of the empirical risk retains an irreducible component of order $\mathcal{O}\big(\frac{1}{N}\mathbb{E}[\text{tr}\,\Sigma_{\text{post}}(G)]\big)$, leading to noisier optimization and slower convergence.

### E.4 Implications and Protocol Choice

Two formal implications follow: (i) under squared loss, random pairing forces the Bayes predictor to the group posterior mean, eliminating useful cell-level signal; (ii) independence between paired samples injects additional, input-agnostic gradient variance tied to $\Sigma_{\text{post}}(G)$. Accordingly, the experimental protocol adopts plate-matched DMSO pairing with within-key averaging, which avoids group-mean degeneration and reduces uninformative gradient noise, yielding a better-conditioned learning problem for perturbation prediction.

## F Ablation Settings and Detailed Results

**Protocol.** All ablations retrain the model under identical training budgets and report on held-out UC/UD splits. Unless otherwise noted, we keep the latent VAE, VP schedule, optimizer, and sampling steps fixed, changing only the factor under test. We report correlations on $\log_2$ FC and $\Delta$, explained variance, and DE-Overlap/Top-$k$ metrics.

### F.1 Unconditional lower bound (No-Cond)

**Setting.** Remove all conditioning paths. **Observation.** Uniform drops across metrics on UC/UD establish a unified lower bound and confirm that modeling state + drug + dose is necessary.

### F.2 Injection strategy: GD-Attn → Concat+MLP

**Setting.** Replace token-level self → cross($c$) → FiLM with input-level concatenation $[z; c]$ followed by MLP. **Observation.** Overall fit and $\Delta$-direction consistency decline, with dose-rank consistency largely unchanged, indicating non-concatenative, selectively coupled injection improves representation and alignment.

### F.3 Drug representation prior: Morgan/SMILES → Drug-ID

**Setting.** Replace structure-informed drug embedding with the ID embedding without structural prior. **Observation.** The largest degradations appear on UD metrics (both direction and magnitude), confirming the necessity of the structure→MoA inductive bias for leave-compound generalization.

### F.4 Guided decomposability: dual-knob CFG → single-channel CFG

**Setting.** Replace decomposable guidance $\hat{\epsilon} = (1 + s_p + s_d)\epsilon_{\text{both}} - s_p\epsilon_{\text{no–pre}} - s_d\epsilon_{\text{no–drug}}$ with a single scalar CFG. **Observation.** Robustness and combinatorial balancing degrade, especially on UD; independent control of state ($s_p$) and drug ($s_d$) is crucial for interpretability and stability.

### F.5 Dose→strength mapping: $s_d(\text{dose}) \rightarrow s_d = s_0$

**Setting.** Replace log-dose sigmoid mapping with a constant drug guidance. **Observation.** Overall fit is similar, but cross-dose rank monotonicity breaks (e.g., Spearman on $\log_2 \text{FC}$), showing that mapping dose to guidance strength is the key mechanism for dose-aware extrapolation.

**Notes on reproducibility.** Hyperparameters are kept identical across compared variants except for the ablated factor.

## G Implementation Details: Tokenization and GD-Attn Injection

### G.1 Tokenization / de-tokenization.

We map the latent vector $z \in \mathbb{R}^{D_z}$ to $M$ tokens (width $d$, so $Md = D_z$) and back via learned linear layers:

$$y = W_{\text{tok}}\, z + b_{\text{tok}}. \tag{26}$$

$$H = \text{reshape}(y) \in \mathbb{R}^{M \times d}, \qquad Md = D_z. \tag{27}$$

$$\tilde{z} = W_{\text{det}}\, \text{vec}(H^{\text{out}}) + b_{\text{det}}. \tag{28}$$

Here $W_{\text{tok}}, b_{\text{tok}}, W_{\text{det}}, b_{\text{det}}$ are learned; $\text{vec}(\cdot)$ flattens tokens.

### G.2 Condition token.

The baseline state and drug–dose are encoded and fused into a compact condition representation

$$c = F\big(z_{\text{pre}},\, \tilde{z}_{\text{drug}}\big) \in \mathbb{R}^{D_c}, \quad z_{\text{pre}} = E(x_{\text{pre}}) \text{ (posterior mean)}. \tag{29}$$

### G.3 GD-Attn block (self → cross with $c$ → FiLM).

Let $H \in \mathbb{R}^{M \times d}$ be tokens and $h$ the number of heads ($d_h = d/h$).

$$Q^{(m)} = HW_Q^{(m)}, \quad K^{(m)} = HW_K^{(m)}, \quad V^{(m)} = HW_V^{(m)},$$
$$A^{(m)} = \text{Softmax}\Big(\tfrac{Q^{(m)}(K^{(m)})^\top}{\sqrt{d_h}}\Big), \quad O^{(m)} = A^{(m)}V^{(m)}, \qquad \tilde{H} = \big[O^{(1)}\|\cdots\|O^{(h)}\big]W_O. \tag{30}$$

Cross-attention with condition token $c$ (projected to $u = W_c c$):

$$Q_c^{(m)} = \tilde{H}\,\tilde{W}_Q^{(m)}, \quad K_c^{(m)} = u\,\tilde{W}_K^{(m)}, \quad V_c^{(m)} = u\,\tilde{W}_V^{(m)},$$

$$A_c^{(m)} = \mathrm{Softmax}\!\left(\frac{Q_c^{(m)}(K_c^{(m)})^\top}{\sqrt{d_h}}\right), \quad O_c^{(m)} = A_c^{(m)}V_c^{(m)}, \qquad \hat{H} = \big[O_c^{(1)}\|\cdots\|O_c^{(h)}\big]\tilde{W}_O. \tag{31}$$

FiLM modulation and residual:

$$\gamma(c) = W_\gamma c + b_\gamma, \quad \beta(c) = W_\beta c + b_\beta, \quad H^{\mathrm{out}} = \mathrm{LN}\big(H + \gamma(c)\odot\hat{H} + \beta(c)\big). \tag{32}$$

**Design note.** Compared with direct concatenation $[H; c]$ at the input, this non-concatenative path avoids gradient-scale blowup and early-training batch-noise amplification, while aligning the condition with the backbone's coordinate system and enabling decomposable guidance at inference.

## H  BIOLOGICAL VERIFICATION

### H.1  PATHWAY ENRICHMENT ANALYSIS (GSEA) WITH ES CURVES AND HIT DISTRIBUTIONS

We performed GSEA on the model-predicted perturbation responses and examined both the running enrichment score (ES) curves and the gene-set hit distributions as a biological validity check. Since these analyses are conducted on well-established canonical pathways, they provide a direct way to assess whether the model captures biologically coherent responses rather than matching expression patterns only at the correlation level.

For known pathways, the predicted ES curves follow the expected trend and directionality observed in the ground-truth rankings. Likewise, the hit distributions show that pathway member genes appear in similar regions of the ranked gene list for both prediction and ground truth. Together, these pathway-level enrichment profiles indicate that the model recovers known biological signaling patterns and produces perturbation predictions that are consistent with established mechanistic responses.

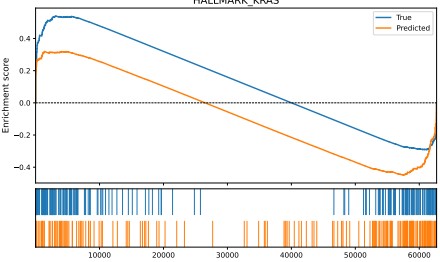 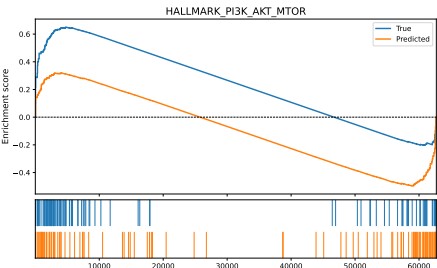

Figure 5: GSEA results on two canonical pathways, HALLMARK_KRAS_SIGNALING_UP (left) and HALLMARK_PI3K_AKT_MTOR_SIGNALING (right). The running enrichment score (ES) curve and the corresponding gene-set hit distribution together provide a biological validity check, as both are evaluated on established canonical pathways. The ES curve reflects how pathway genes accumulate along the ranked list, while the hit plot shows where these genes appear in the ranking, indicating the degree to which the model-predicted perturbation profile aligns with known biological responses.

### H.2  MECHANISM-OF-ACTION (MOA) VALIDATION IN LATENT SPACE

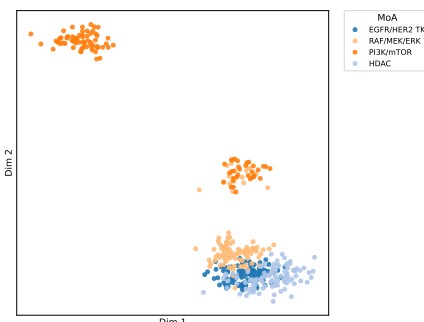

To assess whether scPPDM learns pharmacologically meaningful structure beyond gene-level correlations, we examine the latent representations of drugs used during training and inference. When projected using UMAP, the latent embeddings exhibit clear mechanism-of-action (MoA)–specific organization: compounds with shared or related MoA labels tend to cluster together, while drugs acting through distinct pathways occupy separate regions. This structured organization indicates that scPPDM internalizes mechanistic signals and captures higher-level biochemical relationships, providing additional evidence that the learned latent space reflects real pharmacological structure.

Figure 6: MoA structure in the scPPDM latent space. Drugs form coherent clusters according to their annotated mechanisms of action, indicating that the latent representations learned by scPPDM capture biologically meaningful pharmacological relationships.

## I  COMPLETE COMPARISON TABLES

Table 3: UC test set results. Comparison of baseline models and our method on the UC scenario. Metrics are reported with ↑ indicating that higher is better. **Bold** and underline indicate the best and second method for each metrics, respectively.

| (unseen combo) | Ours | chemCPA | PRnet | STATE | scGPT | Linear | MLP | Perturb | Context |
|---|---|---|---|---|---|---|---|---|---|
| logFC-Pearson ↑ | **0.81** | 0.70 | 0.61 | 0.77 | 0.46 | 0.50 | 0.55 | 0.46 | 0.48 |
| logFC-Spearman ↑ | **0.69** | 0.46 | 0.58 | **0.69** | 0.39 | 0.51 | 0.57 | 0.38 | 0.32 |
| logFC-Pearson(DEG) ↑ | **0.56** | 0.40 | 0.31 | 0.50 | 0.32 | 0.38 | 0.37 | 0.35 | 0.35 |
| logFC-Spearman(DEG) ↑ | **0.49** | 0.38 | 0.28 | 0.48 | 0.23 | 0.34 | 0.23 | 0.33 | 0.32 |
| $\Delta$-Pearson ↑ | 0.55 | **0.59** | 0.35 | 0.50 | 0.39 | 0.48 | 0.44 | 0.21 | 0.25 |
| $\Delta$-Spearman ↑ | **0.51** | 0.44 | 0.27 | 0.50 | 0.35 | 0.41 | 0.36 | 0.25 | 0.24 |
| $\Delta$-Pearson(DEG) ↑ | **0.57** | 0.52 | 0.36 | 0.54 | 0.38 | 0.45 | 0.43 | 0.22 | 0.24 |
| $\Delta$-Spearman(DEG) ↑ | **0.59** | 0.46 | 0.28 | 0.52 | 0.33 | 0.41 | 0.36 | 0.24 | 0.25 |
| DEG-accuracy(top50) ↑ | 0.15 | **0.19** | 0.06 | 0.17 | 0.06 | 0.14 | 0.15 | 0.14 | 0.12 |
| DEG-accuracy(top100) ↑ | 0.19 | **0.21** | 0.09 | 0.19 | 0.08 | 0.15 | 0.16 | 0.16 | 0.13 |
| DEG-accuracy(top200) ↑ | **0.24** | 0.22 | 0.11 | 0.22 | 0.11 | 0.14 | 0.17 | 0.16 | 0.13 |
| DEG-accuracy(top1000) ↑ | **0.39** | 0.28 | 0.21 | 0.32 | 0.19 | 0.19 | 0.18 | 0.17 | 0.15 |
| $E[EV]$ ↑ | 0.62 | 0.62 | 0.48 | **0.64** | 0.37 | 0.22 | 0.37 | 0.46 | 0.40 |
| $E[EV]$(DEG) ↑ | 0.51 | **0.52** | 0.39 | **0.52** | 0.29 | 0.16 | 0.28 | 0.38 | 0.30 |
| $EV$-median ↑ | **0.73** | 0.70 | 0.51 | 0.69 | 0.41 | 0.20 | 0.40 | 0.48 | 0.39 |
| $EV$-median(DEG) ↑ | **0.64** | 0.59 | 0.37 | 0.53 | 0.34 | 0.18 | 0.30 | 0.38 | 0.33 |

Table 4: UD test set results. Evaluation of models on the UD scenario. Perturb, STATE, and scGPT are not included, as they do not directly support UD tasks. Metrics are reported with ↑ indicating that higher is better. **Bold** and underline indicate the best and second method for each metrics, respectively. For consistency, metric definitions follow those used in the UC scenario (Tables 1–2).

| *(unseen drug)* | Ours | Chemcpa | Prnet | Linear | MLP | Context |
|---|---|---|---|---|---|---|
| logFC-Pearson ↑ | **0.67** | 0.42 | 0.59 | 0.43 | 0.40 | 0.31 |
| logFC-Spearman ↑ | **0.58** | 0.44 | 0.48 | 0.49 | 0.36 | 0.32 |
| logFC-Pearson(DEG) ↑ | **0.51** | 0.38 | 0.30 | 0.32 | 0.29 | 0.31 |
| logFC-Spearman(DEG) ↑ | **0.49** | 0.36 | 0.28 | 0.26 | 0.26 | 0.32 |
| $\Delta$-Pearson ↑ | 0.50 | **0.53** | 0.33 | 0.46 | 0.43 | 0.42 |
| $\Delta$-Spearman ↑ | **0.49** | 0.43 | 0.25 | 0.40 | 0.34 | 0.39 |
| $\Delta$-Pearson(DEG) ↑ | **0.53** | 0.48 | 0.34 | 0.40 | 0.41 | 0.43 |
| $\Delta$-Spearman(DEG) ↑ | **0.52** | 0.41 | 0.28 | 0.37 | 0.33 | 0.39 |
| DEG-accuracy(top50) ↑ | 0.14 | **0.17** | 0.06 | 0.11 | 0.12 | 0.14 |
| DEG-accuracy(top100) ↑ | **0.18** | **0.18** | 0.08 | 0.12 | 0.14 | 0.16 |
| DEG-accuracy(top200) ↑ | **0.22** | 0.20 | 0.07 | 0.12 | 0.14 | 0.15 |
| DEG-accuracy(top1000) ↑ | **0.36** | 0.28 | 0.19 | 0.17 | 0.16 | 0.15 |
| $E[EV]$ ↑ | **0.55** | 0.53 | 0.43 | 0.20 | 0.32 | 0.41 |
| $E[EV]$(DEG) ↑ | 0.40 | **0.42** | 0.34 | 0.11 | 0.20 | 0.29 |
| $EV$-median ↑ | **0.66** | 0.59 | 0.42 | 0.18 | 0.32 | 0.40 |
| $EV$-median(DEG) ↑ | **0.56** | 0.47 | 0.31 | 0.11 | 0.25 | 0.30 |

