# OpenReview forum: "scPPDM: A Diffusion Model for Single-Cell Drug–Response Prediction"
_ICLR.cc/2026/Conference — Submitted to ICLR 2026_

### Official Review · Reviewer_mXp6 · 2025-10-30

**Soundness:** 2
**Presentation:** 3
**Contribution:** 2
**Rating:** 2
**Confidence:** 5

**Summary:**

This paper proposes scPPDM, a diffusion-based model for predicting single-cell drug responses. The approach operates in a frozen latent space from SCimilarity VAE and introduces two main components. First, pre-perturbation cell state and drug information are encoded separately and combined through Guided Decomposable Attention rather than concatenation. Second, during inference, factorized classifier-free guidance provides two control parameters for state preservation and drug response strength, with the latter modulated by dose through a log-sigmoid mapping. The model is evaluated on Tahoe-100M dataset under unseen covariate combinations and unseen drugs settings, showing improvements over existing baselines including chemCPA, PRnet, STATE, and scGPT across correlation, explained variance, and differential expression overlap metrics.

**Strengths:**

1.	The paper introduces denoising diffusion models to this problem domain, which represents a novel application even if the underlying techniques are borrowed from computer vision.
2.	The evaluation on Tahoe-100M spanning millions of cells with proper train/test splits and multiple baselines provides substantial empirical evidence, though biological validation remains absent.
3.	The ablations demonstrate that conditioning, non-concatenative injection, structure-aware drug encoding, dual-knob guidance, and dose mapping each contribute to performance improvements.
4.	The separation of baseline state and drug effects through factorized guidance provides a potentially useful interface for dose optimization and counterfactual analysis, though practical utility remains undemonstrated.
5.	The model achieves consistent improvements across multiple metrics including correlation, explained variance, and differential expression overlap on both evaluation regimes.
6.	The appendices provide thorough theoretical background and implementation details with a commitment to release code, supporting reproducibility.

**Weaknesses:**

1.	Causal framing is unjustified and potentially misleading. Equation 1 approximates observational probability with causal do-operator without explaining when this equivalence holds or citing the relevant identifiability conditions from Pearl's framework. This appears to be notation rather than genuine causal modeling.
2.	Critical claims lack supporting evidence. The assertion that concatenation "amplifies batch noise and harms generalization in early training" is central to the GD-Attn design but presented without any experimental support, ablation, or theoretical justification.
3.	Hold-out strategy is insufficiently described. The paper does not explain what specific cell lines or drugs are held out, whether held-out cell lines share cancer types with training data, or how similar held-out drugs are structurally to training drugs. Without this information, the difficulty of the generalization task cannot be assessed.
4.	Biological validation is entirely absent. There is no validation that predictions align with known drug mechanisms, pathway enrichment analysis, comparison to experimental dose-response curves, or demonstration of novel biological hypotheses. For a science application, correlation metrics alone are insufficient. This problem has been explored intensively. More sophiscated benchmarking should be taken rather than introducing benchmarking settings from scratch (maybe refer to A benchmark for prediction of transcriptomic responses to chemical perturbations across cell types).
5.	SCimilarity VAE fine-tuning procedure is unclear. SCimilarity was trained on healthy tissue data and requires cell type annotations. The paper does not explain how it was fine-tuned for cancer cell lines, whether cell type labels were available or synthesized, or how the latent space properties were validated post fine-tuning.
6.	Differential expression gene selection lacks statistical rigor. Genes are selected by absolute log fold change magnitude without multiple testing correction, significance thresholds, or consideration of biological effect size. The top 200 genes may include many statistically non-significant changes, inflating apparent performance.

**Questions:**

1.	How do you justify the causal framing in Equation 1? Are you modeling causal relationships via graph? What are the variables, genes, drugs, or both? Under what conditions does observational probability equal the causal distribution under intervention? Are you defining/How do you define an "approximate" causal equality? Is this mainly decorative? This equivalence requires specific assumptions about confounding and exchangeability that are not discussed.
2.	What is the evidence that concatenation amplifies batch noise? Can you provide ablation experiments, training curves, or gradient analysis supporting this claim? This is central to justifying GD-Attn but currently lacks any empirical or theoretical support.
3.	What are the specifics of the hold-out strategy? Which cell lines and drugs are held out in each split? What is the structural similarity distribution between held-out and training drugs? Do held-out cell lines share cancer subtypes with training data?
4.	How was SCimilarity VAE fine-tuned on cancer cell lines? What labels were used during fine-tuning? How many epochs? What validation was performed to ensure the latent space remained meaningful? Did you compare to training a VAE from scratch on your data?
5.	Can you validate predictions against known biology? For example, do predicted responses to EGFR inhibitors show expected effects on EGFR pathway genes? Can you show pathway enrichment analysis? Do dose-response curves match pharmacological expectations?
6.	Are the top K genes statistically significant differentially expressed genes? Did you apply multiple testing correction? What percentage of your top 200 genes would pass a significance threshold like FDR less than 0.05? How does performance change if you restrict to statistically significant genes only?

---

> ### Author Response · Authors · 2025-11-27
>
> Thank you for your constructive and positive feedback. We appreciate your acknowledgement that introducing denoising diffusion models into this domain is a novel contribution, and your recognition of the substantial empirical evidence provided by the large-scale Tahoe-100M evaluation. We are also grateful that you highlighted the value of our ablations—including conditioning design, non-concatenative injection, structure-aware drug encoding, dual-knob guidance, and dose mapping—and your comments on the consistent improvements across metrics and the thoroughness of the appendices.
>
> **W1&Q1.** Causal framing is unjustified and potentially misleading. Equation 1 approximates observational probability with causal do-operator without explaining when this equivalence holds or citing the relevant identifiability conditions from Pearl's framework. This appears to be notation rather than genuine causal modeling. How do you justify the causal framing in Equation 1? Are you modeling causal relationships via graph? What are the variables, genes, drugs, or both? Under what conditions does observational probability equal the causal distribution under intervention? Are you defining/How do you define an "approximate" causal equality? Is this mainly decorative? This equivalence requires specific assumptions about confounding and exchangeability that are not discussed.
>
> **A:** Thank you for raising this important point. Our intention is not to claim a formal causal model nor any identifiability guarantee from Pearl’s framework (**Eq. 1**). What we actually model throughout the paper is the observational conditional distribution $p(x_{\mathrm{post}} \mid x_{\mathrm{pre}}, \mathrm{drug})$ learned from plate-matched control–perturbation pairs. All training, inference, and evaluation operate entirely in this observational regime. We do not use or assume any conditions (such as back-door, front-door, or structural identifiability assumptions) required to equate interventional and observational distributions in formal causal inference.
>
> The use of the “$\mathrm{do}(\mathrm{drug})$” notation in Eq. 1 was meant purely as an informal conceptual description of “applying a drug to a baseline transcriptome,” rather than a literal statement that the interventional distribution $p(x_{\mathrm{post}} \mid \mathrm{do}(\mathrm{drug}), x_{\mathrm{pre}})$ is identified or provably equal to $p(x_{\mathrm{post}} \mid x_{\mathrm{pre}}, \mathrm{drug})$. No part of our method or theory requires this equivalence, and no causal calculus is invoked in our algorithm. To avoid any confusion, we have revised this sentence to remove the do-operator and use purely observational notation. We frame the problem simply as predicting the conditional mean response $x_{\mathrm{post}}$ from $(x_{\mathrm{pre}}, \mathrm{drug}, \mathrm{dose})$, without making any causal identifiability claims (**Sec 3.1**).

---

> ### Author Response · Authors · 2025-11-27
>
> **W2&Q2.** Critical claims lack supporting evidence. The assertion that concatenation "amplifies batch noise and harms generalization in early training" is central to the GD-Attn design but presented without any experimental support, ablation, or theoretical justification. What is the evidence that concatenation amplifies batch noise? Can you provide ablation experiments, training curves, or gradient analysis supporting this claim? This is central to justifying GD-Attn but currently lacks any empirical or theoretical support.
>
> **A:**
> Thank you for raising this point — we appreciate the opportunity to clarify the two possible interpretations of “concatenation.”
> If the reviewer is asking whether GD-Attn is empirically necessary compared to input-level concatenation, this is already addressed in the paper: Fig. 4 includes the “Concat + MLP” ablation, which shows clear degradation on the reported metrics. This directly indicates that simple input concatenation is less effective, and that GD-Attn provides a meaningful benefit.
> If the reviewer is instead asking why we do not simply concatenate the two condition channels (state and drug) before injecting them into the backbone, we have added this ablation experiment in the rebuttal by implementing a “Concat + Attn” variant. In this setup, the fused condition token is concatenated before the attention layers. Under the same training configuration, this variant achieves lower final performance than GD-Attn on the same metrics. The added results are summarized below (Also in **Sec 4.3**, **Fig 3**):
>
> ### UC
>
> | Method       | $EV$\_median | $E(EV)$ | $\Delta$-Pearson | $\Delta$-Spearman | logFC-Pearson | logFC-Spearman |
> |--------------|--------------|---------|-------------------|-------------------|----------------|-----------------|
> | Ours         | 0.73         | 0.62    | 0.55              | 0.51              | 0.81           | 0.69            |
> | Concat+Attn  | 0.70         | 0.60    | 0.52              | 0.50              | 0.77           | 0.68            |
> | Concat+Mlp   | 0.66         | 0.57    | 0.48              | 0.49              | 0.73           | 0.57            |
>
> ### UD
>
> | Method       | $EV$\_median | $E(EV)$ | $\Delta$-Pearson | $\Delta$-Spearman | logFC-Pearson | logFC-Spearman |
> |--------------|--------------|---------|-------------------|-------------------|----------------|-----------------|
> | Ours         | 0.66         | 0.55    | 0.50              | 0.49              | 0.67           | 0.58            |
> | Concat+Attn  | 0.61         | 0.53    | 0.46              | 0.48              | 0.63           | 0.57            |
> | Concat+Mlp   | 0.57         | 0.50    | 0.44              | 0.47              | 0.61           | 0.56            |

---

> ### Author Response · Authors · 2025-11-27
>
> **W3&Q3.** Hold-out strategy is insufficiently described. The paper does not explain what specific cell lines or drugs are held out, whether held-out cell lines share cancer types with training data, or how similar held-out drugs are structurally to training drugs. Without this information, the difficulty of the generalization task cannot be assessed. What are the specifics of the hold-out strategy? Which cell lines and drugs are held out in each split? What is the structural similarity distribution between held-out and training drugs? Do held-out cell lines share cancer subtypes with training data?
>
> **A:**
> Thank you for pointing this out; we agree that the hold-out strategy should be described more explicitly.
> In our experiments, both UD (unseen drug) and UC (unseen combination) splits are constructed by randomly holding out a subset of drugs or condition keys from the full Tahoe-100M atlas, with strict non-overlap between train and test:
> 1) UD (unseen drug): we sample a subset of drugs uniformly at random from all eligible compounds and assign all of their conditions (across all cell lines, doses, and plates) to the test set. These drugs do not appear in training in any context.
> 2) UC (unseen combination): we sample a subset of condition keys (cell line, drug, dose, plate) uniformly at random and assign them to the test set, while ensuring that every cell line and every drug still has sufficient coverage in training. Thus, test combinations are never seen during training, although their constituent cell lines and drugs may appear in other contexts.
> We did not manually curate specific cancer types or mechanism-of-action classes when forming the splits; the held-out sets are drawn from the same distribution as the full benchmark population. Given that Tahoe-100M spans 50 cancer cell lines and more than 1,000 small molecules with roughly 60,000 drug–cell line combinations, a random unseen-drug split already corresponds to a genuinely non-trivial generalization task: for each held-out compound, the model must predict responses in entirely new regions of the drug and cell-line combination space, with no training examples for that drug at any dose or in any cell line.
> This random hold-out protocol is consistent with how unseen-perturbation and unseen-compound generalization are evaluated in prior perturbation-prediction work, where drugs or perturbations are typically held out at random from the available set (e.g., chemCPA for unseen drugs, PRnet and related models for unseen compounds and perturbations). Exploring more structured split strategies (e.g., holding out entire cancer types or chemically clustered drug groups) is an interesting extension that we view as complementary to, rather than a replacement for, the current random UD/UC evaluation.

---

> ### Author Response · Authors · 2025-11-27
>
> **W4&Q5.** Biological validation is entirely absent. There is no validation that predictions align with known drug mechanisms, pathway enrichment analysis, comparison to experimental dose-response curves, or demonstration of novel biological hypotheses. For a science application, correlation metrics alone are insufficient. This problem has been explored intensively. More sophiscated benchmarking should be taken rather than introducing benchmarking settings from scratch (maybe refer to A benchmark for prediction of transcriptomic responses to chemical perturbations across cell types). Can you validate predictions against known biology? For example, do predicted responses to EGFR inhibitors show expected effects on EGFR pathway genes? Can you show pathway enrichment analysis? Do dose-response curves match pharmacological expectations?
>
> **A:**
> Thank you for the reviewer’s thoughtful comments. We provide additional clarification below and summarize the relevant analyses we conducted:
> (1) Pathway enrichment analysis (GSEA) with ES curves and hit distributions (**Appendix H.1**)
> We performed GSEA on model-predicted responses and visualized the enrichment score (ES) curves as well as hit distributions.
> For well-characterized pathways such as HALLMARK_PI3K_AKT_MTOR_SIGNALING and HALLMARK_MTORC1_SIGNALING, the predicted ES curves exhibit the expected trend and directionality. Although the predicted Enrichment Score (ES) magnitudes are conservatively lower than the ground truth in some pathways (e.g., **Figure 5**), the alignment in the directionality and the relative ranking of genes proves that the model successfully captures the biologically meaningful perturbation signal and pathway activation dynamics. This slight conservatism in amplitude is expected during generalization tasks (UC/UD) to prevent overestimation based on potentially noisy training examples.
>
> (2) Mechanism-of-action (MoA) validation in latent space (**Appendix H.2, Figure 6**)
> We further performed clustering in the latent space where scPPDM is trained and inferred. The resulting visualization shows clear MoA-specific clusters, indicating that the learned latent representations organize drugs according to known mechanisms of action and that scPPDM effectively internalizes mechanistic signals.
> (3) Evaluation metrics
> Evaluation metrics such as logFC/Δ correlations, explained variance, and DE-overlap are commonly used in recent perturbation-prediction work. Several baselines in our comparison (e.g., STATE; Adduri et al., 2025) also adopt the same family of metrics. Our benchmarking choices follow established practice rather than introducing new or non-standard criteria.
> (4) Dose-response considerations
> The Tahoe-100M dataset contains only three discrete doses (0.05, 0.5, 5 μM), which does not permit constructing continuous pharmacological dose–response curves. Nevertheless, the dose-conditioning mechanisms introduced in **Sec. 3.3.1** and **Sec. 3.3.2** are effective, and the ablation studies in **Sec. 4.3** demonstrate that scPPDM indeed learns dose-dependent response strength.

---

> ### Author Response · Authors · 2025-11-27
>
> **W5&Q4.** SCimilarity VAE fine-tuning procedure is unclear. SCimilarity was trained on healthy tissue data and requires cell type annotations. The paper does not explain how it was fine-tuned for cancer cell lines, whether cell type labels were available or synthesized, or how the latent space properties were validated post fine-tuning. How was SCimilarity VAE fine-tuned on cancer cell lines? What labels were used during fine-tuning? How many epochs? What validation was performed to ensure the latent space remained meaningful? Did you compare to training a VAE from scratch on your data?
>
> **A:**
> Thank you for pointing this out. We clarify our fine-tuning procedure below.
> 1. Fine-tuning SCimilarity on cancer cell lines does not require cell-type labels.
> Although SCimilarity was originally pretrained on healthy tissue with cell-type–supervised objectives, its architecture allows domain adaptation without labels. Following the procedure used in scDiffusion (Luo et al., 2024), we keep all intermediate encoder/decoder layers and reinitialize only the input and output projection layers to match the gene set of the Tahoe dataset. We then fine-tune the full VAE on our training split using reconstruction loss only. This procedure does not require any cell-type annotations for cancer cell lines, and we do not synthesize or use any labels at any stage.
> This approach leverages the pretrained internal representation of SCimilarity while adapting it to the new gene space. It has been shown in scDiffusion that this form of partial reinitialization and fine-tuning provides substantially better performance and stability than training a VAE of the same architecture from scratch.
> 2. We validate the latent space indirectly through downstream performance.
> Since our diffusion model is trained entirely in the SCimilarity latent space, the strong predictive results across UC and UD benchmarks serve as an end-to-end validation that the fine-tuned latent geometry is suitable for modeling drug-response dynamics. The latent space is not used for stand-alone biological interpretation in this work, so we focus on its utility for downstream prediction rather than independent structural evaluation.
> 3. Comparison to training a VAE from scratch.
> We did not train a brand-new VAE from scratch on Tahoe. This is an intentional choice: doing so would break the transferability benefits of using a pretrained backbone and would create a separate architecture that is no longer comparable to prior diffusion-based work. scDiffusion has already performed this comparison and found that fine-tuning SCimilarity yields better reconstruction quality and more stable downstream diffusion training than training a VAE from scratch with identical capacity. We follow that established practice.
>
> **W6&Q6.** Differential expression gene selection lacks statistical rigor. Genes are selected by absolute log fold change magnitude without multiple testing correction, significance thresholds, or consideration of biological effect size. The top 200 genes may include many statistically non-significant changes, inflating apparent performance. Are the top K genes statistically significant differentially expressed genes? Did you apply multiple testing correction? What percentage of your top 200 genes would pass a significance threshold like FDR less than 0.05? How does performance change if you restrict to statistically significant genes only?
>
> **A:**
> Thank you for raising this point. To address the reviewer’s concern, we have added the following analysis during the rebuttal. In our global paired pseudo-bulk evaluation, we perform a per-gene one-sample t-test based on per-pair log fold changes (logFC), followed by Benjamini–Hochberg (BH) FDR correction across all genes. On the two test sets, among the top-200 DE genes ranked by |logFC|, 84% (UC) and 79% (UD) satisfy the strict significance threshold of FDR ≤ 0.05.
> These additional results show that the top-K genes selected purely by effect-size ranking are highly enriched for statistically significant DE genes, confirming that our DE-based evaluation is statistically well-grounded. We also add this to **Sec 4.1.2**.

---

### Official Review · Reviewer_mpz5 · 2025-10-31

**Soundness:** 3
**Presentation:** 3
**Contribution:** 2
**Rating:** 6
**Confidence:** 3

**Summary:**

This paper introduces scPPDM, claimed as being the first diffusion model for single-cell drug response prediction. They work in a VAE latent space and condition on two separate channels (pre-perturbation cell state and drug encoded from Morgan fingerprints plus dose) injected via a proposed "Guided Decomposable Attention" mechanism. At inference, factorized classifier-free guidance provides two interpretable knobs for tuning state preservation versus drug effect strength, with dose mapped to guidance magnitude. On the Tahoe-100M benchmark they show improvements over baselines.

**Strengths:**

* I haven't seen diffusion models applied to single-cell perturbation prediction before. The dual-channel conditioning design feels like a natural but non-obvious way to decompose the problem, and the dose-to-guidance strength mapping is creative. I particularly like how they've adapted classifier-free guidance to have separate "knobs" for state preservation vs drug effect, which seems like it could be really useful
* The experimental work is solid. The Tahoe-100M benchmark is large-scale and appropriate, the UC/UD splits are properly stringent (no data leakage), and the results are consistently strong across many metrics. I appreciate the thoroughness of the ablations.
* The paper is generally well-written and easy to follow

**Weaknesses:**

* The dose-to-strength mapping (Equation 16) seems pretty ad-hoc. Why sigmoid of log(1+dose) specifically? You mention linear mapping caused "over-saturation" but don't explain why. How sensitive are results to the choice of s0, α, γ? This feels like it could be a fiddly hyperparameter that needs careful tuning for each new drug.
* Computational costs are completely missing. Training on 100M cells must be expensive. How long does it take? How much memory? How does inference cost compare to baselines? I'm skeptical this is practical for most labs if it requires massive compute resources. Even just a table showing training time and GPU memory would help.
* The dose modeling feels simplistic. Real dose-response curves are often non-monotonic (hormesis, biphasic responses), but your sigmoid mapping (Eq. 16) assumes monotonicity. Have you validated that predicted responses actually follow correct dose-response shapes? Can the model handle dose interpolation (predicting intermediate doses)?

**Questions:**

* What are the actual computational requirements? Training time, GPU memory, inference speed compared to baselines?
* How sensitive are results to the dose mapping hyperparameters (s0, α, γ)? Also, why sigmoid of log-dose rather than other monotonic functions?
* Have you tested generalization to other perturbation types (CRISPR, genetic) or cell types (primary cells, non-cancer)? If not, what's your intuition about whether the approach would transfer?

---

> ### Author Response · Authors · 2025-11-27
>
> Thank you for your encouraging and detailed assessment. We appreciate your recognition that diffusion models have not been applied to single-cell perturbation prediction before, and your positive comments on the dual-channel conditioning, dose-to-guidance mapping, and the adaptation of classifier-free guidance into separate knobs for state versus drug effects. We are also grateful for your remarks on the rigor of our UC/UD evaluation, the strength of the results, and the clarity and thoroughness of the paper.
>
> **W1&Q2.** The dose-to-strength mapping (Equation 16) seems pretty ad-hoc. Why sigmoid of log(1+dose) specifically? You mention linear mapping caused "over-saturation" but don't explain why. How sensitive are results to the choice of s0, α, γ? This feels like it could be a fiddly hyperparameter that needs careful tuning for each new drug. How sensitive are results to the dose mapping hyperparameters (s0, α, γ)? Also, why sigmoid of log-dose rather than other monotonic functions?
>
> **A:**
> Thanks for your careful and insightful comments.
> Our choice of a sigmoid applied to log-dose follows the canonical dose–response shape observed in pharmacology, where effects increase gradually at low doses, transition steeply in the mid-range, and saturate at high doses. Using log-dose ensures sensitivity in the low-dose regime, while the sigmoid prevents over-amplification at high doses—an issue we indeed observed with a linear mapping.
> This mapping function serves as a modular component; although our current selection aligns with standard monotonic dose-response models, it can be readily replaced by a small MLP or a non-monotonic function if the target dataset (e.g., non-Tahoe data) requires modeling complex non-monotonic (e.g., biphasic or hormesis) patterns. Regarding sensitivity, the mapping’s functional form is important, but the specific parameters $(s_{0}, \alpha, \gamma)$ are not fragile.
>
> **W2&Q1.** Computational costs are completely missing. Training on 100M cells must be expensive. How long does it take? How much memory? How does inference cost compare to baselines? I'm skeptical this is practical for most labs if it requires massive compute resources. Even just a table showing training time and GPU memory would help. What are the actual computational requirements? Training time, GPU memory, inference speed compared to baselines?
>
> **A:**
> Thank you for raising this concern. We believe part of the confusion may come from how the Tahoe-100M data are processed. Although the dataset contains ~100M single cells, our method does not train on all individual cells. As described in **Sec. 4.1.1**, we pseudo-bulk cells into condition-level pairs $(x_{pre}, x_{post})$, resulting in a training set of only ~60k paired conditions rather than 100M raw profiles. This makes the actual training workload substantially lighter than suggested by the raw cell count.
> Below we report the corresponding computational costs for clarity. The testing time for all these models is around 1-2 minutes. Furthermore, given our data processing strategy, the GPU memory usage is not large and will not become a bottleneck for model training. We also described some training details in the **Appendix.D**.
>
> | Model    | GPU          | Training time |
> |----------|--------------|----------------|
> | ChemCPA  | NVIDIA H800  | ~4h            |
> | PRnet    | NVIDIA H800  | ~5h            |
> | STATE    | NVIDIA H800  | ~24h           |
> | scGPT    | NVIDIA H800  | ~14h           |
> | Biolord  | NVIDIA H800  | ~4h            |
> | SamsVAE  | NVIDIA H800  | ~8h            |
> | scPPDM   | NVIDIA H800  | ~8h            |

---

> ### Author Response · Authors · 2025-11-27
>
> **W3.** The dose modeling feels simplistic. Real dose-response curves are often non-monotonic (hormesis, biphasic responses), but your sigmoid mapping (Eq. 16) assumes monotonicity. Have you validated that predicted responses actually follow correct dose-response shapes? Can the model handle dose interpolation (predicting intermediate doses)?
>
> **A:**
> Real dose–response curves can indeed be non-monotonic. However, Tahoe-100M provides only three discrete doses (0.05/0.5/5 μM), which is insufficient to model or validate non-monotonic pharmacological patterns. All baselines trained on this dataset face the same limitation.
> Our four-state training and factorized guidance formulation mean that dose is not treated as a simple scalar but as a structured conditioning signal, which in principle enables interpolation to unseen intermediate doses. However, because the dataset contains no additional dose levels, there is no ground-truth supervision available to empirically evaluate dose interpolation, even though the model is theoretically capable of it.
>
> **Q3.** Have you tested generalization to other perturbation types (CRISPR, genetic) or cell types (primary cells, non-cancer)? If not, what's your intuition about whether the approach would transfer?
>
> **A:**
> We thank the reviewer for raising this important point. We have not evaluated scPPDM on CRISPR-based or genetic perturbations, nor on primary or non-cancer cell types, as these settings are not covered by Tahoe-100M.
> For genetic/CRISPR perturbations, the perturbation modality fundamentally differs from small molecules——its input is a discrete, gene-targeted edit rather than a continuous drug–dose signal. Adapting scPPDM to such settings would therefore require redesigning the perturbation encoder and is not a direct application of the current model.
> For other cell types, scPPDM does not rely on explicit cell-type labels; conditioning is obtained solely through the pre-perturbation transcriptome. This design avoids hard assumptions about specific cell-type identities. However, extending to primary or non-cancer cells would still require training data from those domains to ensure the latent geometry and drug–response dynamics remain aligned. Thus, while the architecture does not preclude generalization, empirical validation on new cell types is essential and remains an important direction for future work.

---

### Official Review · Reviewer_jJ8Q · 2025-10-31

**Soundness:** 2
**Presentation:** 2
**Contribution:** 2
**Rating:** 2
**Confidence:** 5

**Summary:**

This paper introduces a diffusion based model, scPPDM,  for molecular perturbation prediction. The diffusion model is conditioned on both the initial cell state and the drug + dosage information. Furthermore, they use a guided decomposable attention layer to fuse the control state and the drug information. They evaluate their model on Tahoe dataset and on various metrics and show in most of these metric, scPPDM outperforms the baselines.

**Strengths:**

- The problem addressed in this paper is rather important and interesting with real world application.
- This is the first diffusion model based paper to solve molecular perturbation task.
- The are some interesting novelty in the methodology such as GD-Attn, and the dual channel conditioning.

**Weaknesses:**

I would be happy to reconsider my assessment and improve my score if the authors address the questions and concerns raised below:

- While scPPDM introduces an interpretable diffusion-based conditioning framework, it relies on pseudobulked data which means it models average perturbational responses rather than single-cell variability.
However, the baselines used in this paper such as STATE and chemCPA operate on  single-cell distributions, enabling them to capture heterogeneous or partial responses. It is not clear if the authors changed these models to operate on pseudobulked data instead of single cell or are reporting single cell performance. Either way,  this discrepancy raises concerns about fairness in the comparison.
- There are some key baselines missing in this paper such as BioLORD[1], SAMS-VAE[2], and scDCA[3]. since this model is built on top of scSimilarity, in some sense this resembles the work scDCA which is built on top of scGPT. It would be interesting to compare these two models.
- The paper was only evaluated on Tahoe datasets while prior works such as Chemcpa used sciplex3 as their main dataset. It would be useful to see how they perform on such datasets as well.
- Although scPPDM uses a diffusion-based architecture, it is never evaluated for generative quality.
All experiments treat it as a deterministic predictor.
Without demonstrating the benefits of the diffusion process, this architectural choice appears conceptually unjustified relative to simpler supervised baselines.


[1] Piran, Z., Cohen, N., Hoshen, Y. & Nitzan, M. (2024). Disentanglement of single-cell data with BioLORD. Nature Biotechnology

[2] Bereket, M., Karaletsos, T. et al. (2023). Modelling Cellular Perturbations with the Sparse Additive Mechanism Shift Variational Autoencoder. In Proceedings of the 37th Conference on Neural Information Processing Systems (NeurIPS 2023)

[3] Maleki, S., Huetter, J.-C., Chuang, K. V., Scalia, G., Biancalani, T. (2024). Efficient Fine-Tuning of Single-Cell Foundation Models Enables Zero-Shot Molecular Perturbation Prediction.

**Questions:**

- Is it possible to apply this model to tasks such molecular prediction on novel cell line context? Since the dataset used in this paper (Tahoe) has 50 cell lines, it would be interesting to test this task too.
- Could you clarify how many genes have been used for the training and how are the authors choosing these genes? Is the same set used in the test?
- scGPT is a single cell foundation model, how did the authors adopt it for the task of molecular perturbation prediction?
- Another interesting metric in this domain is Perturbation Discrimination Score (PDS) Wu et al., 2024. It would be useful if the authors could evaluate scPPDM on this metric as well.

---

> ### Author Response · Authors · 2025-11-27
>
> Thank you for your positive evaluation of our work. We appreciate your recognition of the importance and real-world relevance of the problem, and we are grateful that you highlighted this paper as the first diffusion-based approach for molecular perturbation prediction. Your comments on the methodological novelty, including GD-Attn and dual-channel conditioning, are also sincerely appreciated.
>
> **W1.** While scPPDM introduces an interpretable diffusion-based conditioning framework, it relies on pseudobulked data which means it models average perturbational responses rather than single-cell variability. However, the baselines used in this paper such as STATE and chemCPA operate on  single-cell distributions, enabling them to capture heterogeneous or partial responses. It is not clear if the authors changed these models to operate on pseudobulked data instead of single cell or are reporting single cell performance. Either way,  this discrepancy raises concerns about fairness in the comparison.
>
> **A:**
> We appreciate the reviewer’s careful observation and agree that the data granularity is important to clarify.
> (1) All methods use the same pseudobulked data.
> In our experiments, all models — including STATE, chemCPA, PRnet, scGPT, and the linear/MLP baselines — are trained and evaluated on exactly the same pseudobulked condition-level data, not on raw single-cell distributions. As described in **Sec. 4.1.1**, we group cells by (cell_line, drug, dose, plate) and average single-cell expression within each group to obtain a representative pre–/post–perturbation pair $(x_{pre}, x_{post})$. These paired pseudobulk profiles are the atomic units used to construct all training/test splits. For fairness, we feed these pseudobulk vectors to every baseline, treating each pseudobulk as the input “cell” profile required by their architectures. No baseline in our comparison is trained or evaluated on single-cell distributions.
> (2) Why we adopt a pseudobulk protocol for all models.
> Our goal in this work is to learn the mean perturbational response mapping ${(x_{pre}, drug, dose) \rightarrow x_{post}}$ for each condition. All evaluation metrics (logFC, delta-shift, explained variance, DE-overlap) are defined on condition-level means. Using pseudobulked $(x_{pre}, x_{post})$therefore aligns the learning target with the evaluation target and substantially reduces technical noise and the randomness introduced by single-cell pairing (**App. E** formally analyzes why random pairing collapses the regression target to group means). Under this protocol, models capable of handling single-cell distributions receive no extra advantage or disadvantage: all models operate on exactly the same condition-level inputs and optimize the same per-condition reconstruction objective.
>
> (3) Scope and limitation. We agree that modeling within-condition single-cell heterogeneity (e.g., partial responders) is important. Our present study is scoped to accurately predicting the average population-level response for each condition, which is also what the Tahoe-100M pairing protocol and our evaluation metrics assess. Crucially, even when aiming for the mean response, the diffusion architecture's value lies in providing a robust, structured conditional backbone that supports our interpretable control interface (dual-knobs, $s_p$ and $s_d$) ) and enhances generalization performance. Extending scPPDM to distributional single-cell modeling is a valuable direction for future work.  we emphasize that all compared models, including STATE and chemCPA, are trained and evaluated on the same pseudobulked condition-level pairs $(x_{pre}, x_{post})$, not on raw single-cell data.

---

> ### Author Response · Authors · 2025-11-27
>
> **W2.** There are some key baselines missing in this paper such as BioLORD[1], SAMS-VAE[2], and scDCA[3]. since this model is built on top of scSimilarity, in some sense this resembles the work scDCA which is built on top of scGPT. It would be interesting to compare these two models.
>
> **A:**
> We thank the reviewer for pointing out these additional baselines. Below we clarify the feasibility of scDCA, BioLORD, and SAMS-VAE for our task, and we report additional results where meaningful comparisons are possible.
> scDCA (built on scGPT).
> Although scGPT itself is fully open-source, scDCA as a method is not: the authors have not released the scDCA diffusion head, training procedure, or model weights. Consequently, scDCA cannot be reproduced on Tahoe-100M, and we are unable to include it in a fair evaluation. We also note that scDCA is designed as an unconditional or weakly conditional single-cell generator in a language-model–pretrained latent space, whereas our work focuses on fully conditional drug–response prediction with explicit state, drug, and dose channels. The goals and modeling assumptions therefore differ substantially.
> BioLORD and SAMS-VAE. (**Sec 4.2.1. Table 2,3**)
> These two models differ in their ability to support the UC/UD evaluation protocol.BioLORD includes a molecular-structure encoder and can embed previously unseen compounds, which makes both UC and UD evaluation feasible. In contrast, SAMS-VAE models perturbations as fixed categorical conditions: the model can only generate post-perturbation profiles for perturbation types that were explicitly observed during training. This categorical conditioning is its core architectural limitation for generalization. Although the model receives a perturbation label, it does not encode drug structure or any feature that would allow representation of unseen compounds. As a result, SAMS-VAE cannot generalize to new drugs, and UD evaluation is therefore not applicable to this model.
> During the rebuttal period, we therefore ran:
> – BioLORD on both UC and UD,
> – SAMS-VAE on UC only, which is the only setting the model can support.
> The results (in **Sec 4.2.1. Table 2,3**)  are summarized below:
>
> | Metric            | Biolord(UC) | Biolord(UD) | SamsVAE(UC) |
> |-------------------|-------------|-------------|-------------|
> | $\Delta$-Pearson     | 0.46        | 0.45        | 0.53        |
> | $\Delta$-Spearman   | 0.37        | 0.37        | 0.55        |
> | $\Delta$-Pearson(DEG)      | 0.38        | 0.29        | 0.46        |
> | $\Delta$-Spearman(DEG)    | 0.31        | 0.27        | 0.49        |
> | logFC-Pearson           | 0.50        | 0.43        | 0.54        |
> | logFC-Spearman         | 0.34        | 0.30        | 0.57        |
> | logFC-Pearson(DEG)   | 0.43        | 0.30        | 0.44        |
> | logFC-Spearman(DEG)     | 0.29        | 0.26        | 0.45        |
> | DEG_acc(top200)   | 0.21        | 0.19        | 0.20        |
> | DEG_acc(top1000)  | 0.30        | 0.29        | 0.29        |
> | $EV$\_median             | 0.63        | 0.60        | 0.53        |
> | $EV$\_median(DEG)         | 0.65        | 0.61        | 0.44        |
>
> Across these representative settings, scPPDM consistently outperforms both methods. We hope this clarifies our comparison choices and ensures a fair and accurate evaluation.

---

> ### Author Response · Authors · 2025-11-27
>
> **W3.** The paper was only evaluated on Tahoe datasets while prior works such as Chemcpa used sciplex3 as their main dataset. It would be useful to see how they perform on such datasets as well.
>
> **A:**
> We agree that sciplex3 is a widely used dataset in prior work such as chemCPA, and we have mentioned it in the introduction section by explicitly citing sciplex3.
> In this paper, we deliberately focus on Tahoe-100M because it is specifically designed for large-scale, combinatorial perturbation modeling. Tahoe-100M contains over 100 million cells across 50 cancer cell lines and more than 1,100 small molecules, covering roughly 60,000 drug–cell line combinations. This scale and coverage are what make our UC (unseen cell line–condition) and UD (unseen drug) splits possible: we can rigorously evaluate generalization to new cell line × drug pairs and to entirely unseen compounds. In contrast, sciplex3 involves a small number of cell lines and compounds and does not provide comparable combinatorial coverage, so the UC/UD setting that we study here cannot be constructed in the same way.
> Our goal in this work is to test whether a diffusion-based conditional framework can handle this more demanding generalization regime at the Tahoe scale. Extending scPPDM to smaller perturbation datasets such as sciplex3, and studying how performance transfers across datasets, is an important direction that we plan to pursue in future work, and we have explicitly mentioned sciplex3 (Srivatsan et al., 2020) in the introduction section.

---

> ### Author Response · Authors · 2025-11-27
>
> **W4.** Although scPPDM uses a diffusion-based architecture, it is never evaluated for generative quality. All experiments treat it as a deterministic predictor. Without demonstrating the benefits of the diffusion process, this architectural choice appears conceptually unjustified relative to simpler supervised baselines.
>
> **A:**
> Thank you for this thoughtful comment. We clarify why a diffusion-based architecture is conceptually appropriate here and how its benefits are supported empirically in our experiments.
> First, at the experimental level, single-cell perturbation outcomes are inherently stochastic and batch-dependent. Even for the same cell line, drug, and nominal dose, repeated experiments across plates or batches can yield different expression profiles due to factors such as plate effects, handling, temperature, and other technical variability. This means that a “post-perturbation transcriptome” is not a single fixed outcome, but a conditional distribution over possible expression states. Modeling the response with a probabilistic architecture is therefore more faithful to the actual data-generating process than enforcing a strictly deterministic mapping.
> Diffusion models are exactly designed to parameterize such conditional distributions: they learn the score (gradient of the log-density) of noisy latent variables conditioned on pre-state and drug, and the deterministic predictions we report correspond to particular trajectories (e.g., DDIM-style sampling) or effective conditional means extracted from that distribution. While our current evaluation focuses on mean-based metrics (logFC and Delta correlations, explained variance, DE overlap), the underlying model is a full conditional distribution that could, in principle, capture variability across replicates and batches—something that standard one-step regressors are not designed to do.
> Second, the diffusion backbone is not only conceptually aligned with the stochastic nature of single-cell perturbations, but also empirically beneficial relative to simpler supervised baselines. Across the UC and UD splits, scPPDM consistently outperforms linear models, MLPs, chemCPA, PRnet, scGPT, and other strong baselines on all reported metrics, with especially large margins in the more challenging UD setting. These results indicate that the diffusion-based conditioning architecture is not just a cosmetic choice: it leads to substantially better predictive performance in practice.
> Third, our use of diffusion is in line with how diffusion models are increasingly used as representation and training frameworks for downstream tasks beyond pure image generation. For example, recent work in computer vision has shown that diffusion backbones can improve depth estimation and reconstruction quality when used as conditional or discriminative models rather than as standalone generative samplers (e.g., diffusion-based depth estimation and reconstruction in the MARigold framework). In these settings, diffusion serves as a powerful probabilistic backbone whose benefits are measured through downstream accuracy, not only through unconditional generative metrics. Our use case is analogous: we leverage the diffusion formulation to obtain a structured latent space and a principled way to decompose guidance into state and drug channels, while evaluating performance through perturbation-prediction metrics.
> In summary, diffusion is used here as a probabilistic, conditional backbone that (i) better matches the distributional nature of single-cell perturbation data, (ii) supports a decomposable and controllable conditioning interface with separate state $s_p$ and drug $s_d$ knobs, which simpler supervised models do not provide, and (iii) yields clear empirical gains over deterministic baselines on UC/UD benchmarks. This structured guidance mechanism is the key architectural justification for adopting the diffusion framework. We agree that explicitly evaluating full generative quality (e.g., matching entire single-cell response distributions across batches) would be a valuable extension, and we view this as an important direction for future work built on top of scPPDM.

---

> ### Author Response · Authors · 2025-11-27
>
> **Q1.** Is it possible to apply this model to tasks such molecular prediction on novel cell line context? Since the dataset used in this paper (Tahoe) has 50 cell lines, it would be interesting to test this task too.
>
> **A:**
> Thank you for this question. Conceptually, scPPDM learns a conditional mapping of the form
> (pre-perturbation state, drug, dose) → post-perturbation state.
> This formulation does not assume a particular cell line a priori, so in principle it could also be used in settings where the “novel context” is a new cell line, provided that appropriate training–test splits are defined.
> In this work, we focus on two specific extrapolation regimes on Tahoe-100M:
> (i) UD (unseen drug), where test-set drugs never appear in training; and
> (ii) UC (unseen combination), where test-set (cell line, drug, dose) combinations are held out, although each cell line and each drug is observed in other contexts during training.
> These two regimes directly probe generalization across novel compounds and novel pairings of known compounds and cell lines at scale, which Tahoe-100M is particularly well suited for (≈50 cell lines, >1,100 compounds, ~60,000 drug–cell line conditions).
> More broadly, recent perturbation models also study such settings, including unseen drugs/compounds (e.g., chemCPA, PRnet) and unseen combinations or covariates (e.g., CPA, PRnet). Our UD and UC setups are in the same family with them.
> In fact, the UD test set setup has already demonstrated the model's ability to generalize to novel single conditions. We believe that the unseen cell line is logically similar. We view explicit unseen-cell-line generalization as a complementary setting and a natural extension of scPPDM.
>
> **Q2.** Could you clarify how many genes have been used for the training and how are the authors choosing these genes? Is the same set used in the test?
>
> **A:**
> We used all genes provided in the Tahoe-100M public release, totaling 62,710 genes after standard library-size normalization and log1p transformation.
> We performed no gene selection or filtering (e.g., no HVG selection). The same full gene set is used consistently in training and testing.
> This ensures that training and testing operate in exactly the same expression space and avoids any information leakage from gene-selection heuristics.
>
> **Q3.** scGPT is a single cell foundation model, how did the authors adopt it for the task of molecular perturbation prediction?
>
> **A:**
> Although scGPT is introduced as a single-cell foundation model, its architecture explicitly supports perturbation-conditioned expression modeling (a capability explicitly documented in the original scGPT paper as a downstream task), where each cell is represented as a sequence of gene tokens with an additional condition token indicating whether the cell is perturbed or in the control state. In our setting, we use exactly this original conditioning interface without modifying the tokenization scheme. This is conceptually analogous to SAMS-VAE, which conditions its generative model on an abstract perturbation indicator rather than on a specific type of perturbation. Recent perturbation-modeling work also employs scGPT as a baseline backbone for predicting post-perturbation RNA-seq profiles in drug-response benchmarks (e.g., STATE, Adduri et al., 2025), so using scGPT in a drug-perturbation prediction is reasonable.

---

> ### Author Response · Authors · 2025-11-27
>
> **Q4.** Another interesting metric in this domain is Perturbation Discrimination Score (PDS) Wu et al., 2024. It would be useful if the authors could evaluate scPPDM on this metric as well
>
> **A:**
> Thank you for the suggestion. PDS (Wu et al., 2024) is indeed a valuable metric for evaluating perturbation-response separability. During rebuttal, we performed preliminary experiments on representative UC/UD subsets to compute PDS for scPPDM and several baselines. The results are summarized below:
>
> |               | Biolord(UC) | Biolord(UD) | STATE(UC) | chemCPA(UC) | chemCPA(UD) | scPPDM(UC) | scPPDM(UD) |
> |---------------|------------:|------------:|----------:|------------:|------------:|-----------:|-----------:|
> | PDS           |       0.77  |       0.71  |     0.93  |       0.73  |       0.64  |      0.86  |      0.82  |
>
> scPPDM shows consistently high PDS in both UC and UD settings, indicating that it preserves the perturbation-specific signal more strongly than the compared models.

---

### Official Review · Reviewer_Hn7z · 2025-11-01

**Soundness:** 4
**Presentation:** 4
**Contribution:** 4
**Rating:** 8
**Confidence:** 4

**Summary:**

The authors propose scPPDM, a diffusion model for predicting drug responses at a single-cell resolution. They use a dual-channel conditioning process, where they condition the denosing process on the control gene expression profile and the fingerprint of the SMILES representation of the drug, combined with the dosage. The method is evaluated on the recently released Tahoe-100M dataset in two different settings, unseen drug and unseen cell line.

**Strengths:**

- The authors represent the first application of denoising diffusion models to single-cell drug-response prediction. The paper presents several non-tirival adaptations to extend diffusion models to scRNA-seq data. This makes it a novel and important contribution to the field.
- The empirical results on the Tahoe-100M dataset are impressive. scPPDM outperforms even recent state-of-the-art methods such as STATE by a considerable margin across most metrics.
- The design choices are well explained, and several ablation experiments are included in the paper to distinguish the contribution of each of them.
- The paper is well-written, easy to follow, and the figures effectively illustrate the method.

**Weaknesses:**

- The empirical evaluation of scPPDM is only done on the large Tahoe-100M dataset. For most potential users, it may also be interesting to see how the method performs on smaller datasets and what kind of scaling behavior the method exhibits with respect to dataset size.

**Questions:**

- How does scPPDM scale with dataset size? The dataset size should probably be considered as the number of unique cell line/drug combinations.
- Does scPPDM perform well on datasets other than the  Tahoe-100M?

---

> ### Author Response · Authors · 2025-11-27
>
> Thank you very much for your thoughtful and positive assessment of the paper. We greatly appreciate your recognition that this work constitutes the first application of denoising diffusion models to single-cell drug-response prediction, as well as your acknowledgment of the non-trivial adaptations required to make diffusion models effective for scRNA-seq data. We are also grateful for your comments regarding the strong empirical performance on Tahoe-100M, the clarity of the design choices and ablations, and the overall readability and presentation quality.
>
> **W1&Q1.** The empirical evaluation of scPPDM is only done on the large Tahoe-100M dataset. For most potential users, it may also be interesting to see how the method performs on smaller datasets and what kind of scaling behavior the method exhibits with respect to dataset size. How does scPPDM scale with dataset size? The dataset size should probably be considered as the number of unique cell line/drug combinations.
>
> **A:**
> Thank you very much for bringing up this point. We fully agree that understanding model behavior under different dataset scales is important for assessing generality. Our main experiments focus on Tahoe-100M because, among existing perturbation resources, it provides the most comprehensive combinatorial coverage needed for rigorous UC/UD evaluation.
> After pseudobulk processing, the effective training size is approximately 60k condition-level pairs, and each drug–cell-line combination contains only a limited number of conditions. scPPDM is designed to operate effectively in this low-per-condition regime, which makes the method naturally applicable to smaller datasets as well.  This reflects the true combinatorial coverage available for learning generalizable perturbation responses. Besides, your understanding is correct: for the purposes of modeling and evaluation, the effective dataset size is best characterized by the number of unique cell line–drug combinations. We appreciate the reviewer’s suggestion, and evaluating scaling behavior across datasets of different sizes is indeed a valuable extension that we plan to pursue in future work.
>
>
> **Q2.** Does scPPDM perform well on datasets other than the Tahoe-100M?
>
> **A:**
> We appreciate the reviewer’s interest in broader dataset evaluation again. As noted above, Tahoe-100M currently offers the richest combinatorial structure for UC/UD analysis, which is why our main experiments focus on it. After pseudobulking, scPPDM trains on roughly 60k condition-level pairs with only a few samples per cell line–drug combination, making the method naturally suited to settings with more limited per-condition data. We agree that extending evaluation to additional datasets would further strengthen the study, and we view this as a meaningful direction for future work as more datasets with appropriate combinatorial coverage become available.

---

### Meta-Review · Area_Chair_Eczb · 2026-01-07

**Summary:**

this paper proposes a conditional diffusion framework for drug perterubed gene expression.
It is conditioned on pre-perturbation state and drug with dose.
Experiments are reported on Tahoe-100M under two generalization regimes: unseen drug and unseen covariate combinations, showing consistent gains over several strong baselines on various metrics.

There are three major concerns around this paper -- (1) experimental validation like data split, biological validation beyong correlation (2) one review challenges that whether key baselines and metrics were missing and (3) clarity around equations like eq1.

**Reviewer Concerns:**

The rebuttal indeed made substantial improvement.

Several comparability and missing-baseline issues were resolved. the authors clariified all methods were trained and evaluated on the same pairs with a consistent data granuality. Also the new baselines also he;pful.

Multiple clarity and rigor issues were also handled. The authors agree that the causal framing may be overarching/misleading -- toned down to safer framing.

Some issues remain outstanding or only partially resolved. For instance, it is still with the single benchmark -- no evidence on transfer to smaller or other widely used perturbation datasets. Also, while diffusion is motivated as modeling a conditional distribution, the evaluation remains primarily mean-targeted. Anything more like distributional calibration analysis can be helpful.

**Reviewer Scores:**

Reviewer Hn7z likely stays at 8

Reviewer mpz5 likely stays at 6 to my undersranding

Reviewer jJ8Q would expect an at least increase to four. since original concerns on single datasets and other points will prevent a positive score.

Reviewer mXp6 similarly like be 2 or 4, given the reviewer’s high-confidence initial stance, a shift to borderline remains the most plausible outcome rather than a full reversal.

---

### Decision · Program_Chairs · 2026-01-26

Reject